

# AIRTRAC v2.0: a Lagrangian aerosol tagging submodel for the analysis of aviation SO₄ transport patterns

Jin Maruhashi[1,a], Mattia Righi[2], Monica Sharma[1,2], Johannes Hendricks[2], Patrick Jöckel[2], Volker Grewe[1,2], Irene C. Dedoussi[1,3]

[1]Delft University of Technology, Faculty of Aerospace Engineering, Operations and Environment, Delft, The Netherlands
[a]now at: Department of Civil and Environmental Engineering, Imperial College London, London, United Kingdom
[2]Deutsches Zentrum für Luft- und Raumfahrt (DLR), Institut für Physik der Atmosphäre, Oberpfaffenhofen, Germany
[3]Department of Engineering, University of Cambridge, Cambridge, United Kingdom

*Correspondence to*: Irene C. Dedoussi (icd23@cam.ac.uk)

**Abstract.** Aviation-induced aerosols, particularly composed of sulfate ($SO_4$), can interact with liquid clouds by enhancing their reflectivity and lifetime, thereby exerting a cooling effect. The magnitude of these interactions, however, remains highly uncertain and may even offset the combined warming from aviation's other climate forcers depending on spatiotemporal factors such as emission altitude and season. Here, we introduce AIRTRAC v2.0, the latest advancement of the Lagrangian tagging submodel within the Modular Earth Submodel System (MESSy), and the first submodel to provide aviation-specific

sulfate tagging in this framework. AIRTRAC contributes to lowering uncertainty by tracking global contributions of aviation-emitted sulfur dioxide ($SO_2$) and sulfuric acid ($H_2SO_4$) to $SO_4$ formation. Using a sulfur-species tagging approach for $SO_2$, $H_2SO_4$ and $SO_4$, it enables the characterization of transport patterns and highlights atmospheric regions with enhanced potential for aerosol–cloud interactions. In contrast to some of the existing sulfate tagging models, AIRTRAC considers a full range of microphysical processes along trajectories. To investigate sulfate transport from aviation, two global simulations were

performed for January–March and July–September 2015, using pulse emissions of $SO_2$ and $H_2SO_4$ distributed across a cruise altitude of 240 hPa (~10.6 km) based on the aviation $SO_2$ inventory of the Coupled Model Intercomparison Project Phase 6 (CMIP6). Comparisons of AIRTRAC-derived $SO_4$ distributions with perturbation-based simulations under analogous conditions show reasonable agreement. Using AIRTRAC v2.0, we estimate median $SO_2$ and $SO_4$ lifetimes of 22 d and 2.1 months, respectively, in northern winter, and 14 d and 2.2 months in summer, consistent with volcanic eruption modeling and

observational benchmarks involving high-altitude $SO_2$ injection. The median $SO_4$ production efficiency during summer was found to be statistically significantly larger by 144% compared to winter, due to a more efficient oxidation of $SO_2$. Large-scale circulation patterns may contribute to enhancing $SO_4$ lifetimes, especially when injected in the Tropics, where emissions could ascend into the stratosphere, past 100 hPa (~16 km). AIRTRAC v2.0 currently excludes $SO_2$ oxidation from aviation nitrogen oxides ($NO_x$) and does not tag other species such as black carbon. Owing to its flexible design, however, the approach can be

readily extended to additional aerosols. Overall, AIRTRAC v2.0 offers the novel capability to track the atmospheric transport of aviation-emitted $SO_2$, $H_2SO_4$ and $SO_4$, providing critical insights into one of aviation's most uncertain climate impacts.



# 1 Introduction

Recent estimates suggest that in 2018, aviation accounted for approximately 2% of the total anthropogenic radiative forcing (RF) from carbon dioxide ($CO_2$; Klower et al., 2021) emissions and about 3.5% or ~150 mWm$^{-2}$ (70-229 mWm$^{-2}$ for a 90% confidence interval) of all anthropogenic warming when additional non-$CO_2$ effects (Lee et al., 2021) are also contemplated. The latter estimate, however, carries significant uncertainties due to the intricate interactions and trade-offs between various climate forcers, including nitrogen oxides ($NO_x$), water vapor ($H_2O$) and persistent contrails. Additionally, this estimate includes warming and cooling from direct absorption and scattering effects of soot (black carbon (BC) and organic carbon (OC)) and sulfate ($SO_4$) aerosols, respectively. However, it does not consider any indirect RF contributions from their interactions with clouds. Furthermore, Lee et al. (2021) do not include the potential influence of aviation $NO_x$ on $SO_4$ and nitrate ($NO_3$) aerosol formation, both of which may contribute with substantial cooling (Terrenoire et al., 2022).

Notably, the indirect cooling effect of $SO_4$ aerosols alone could reduce aviation's net radiative forcing considerably. Some estimates indicate that the absolute value of this cooling effect may range from 17 to 160 mWm$^{-2}$ (Gettelman and Chen, 2013; Kapadia et al., 2016; Fig. 5 in Lee et al., 2021), a magnitude comparable to aviation's largest global mean warming contribution from aircraft-induced contrail cirrus—estimated at 111 [33, 189] mW m$^{-2}$ for 2018 (Lee et al., 2021) and 62 mW m$^{-2}$ for 2019 (Teoh et al., 2019). Another study broadens this analysis by including both direct and indirect aerosol-induced $SO_4$ forcings, suggesting that their combined cooling effect could again surpass the warming from aviation-induced contrail cirrus, although indirect aerosol effects are typically more dominant (Righi et al., 2013). These combined aerosol forcings raise the possibility of a near-zero or even net negative radiative forcing (RF) from kerosene-powered aircraft. However, this outcome remains highly uncertain due to inherent complexities in quantifying indirect aerosol effects. More robust estimates from independent modeling efforts, supported by a better understanding of critical processes such as pollutant transport patterns, are essential. Without clearer insights into these climate forcers, the effective development and comprehensive evaluation of mitigation strategies will remain challenging.

Aerosols are collections of solid or liquid particles suspended in the atmosphere, with sizes ranging from 0.001 µm to 100 µm (Petzold and Kärcher, 2012; Brasseur and Jacob, 2017). Among the aerosols resulting from aviation are soot and sulfate. Soot is produced from the incomplete combustion of aromatic compounds in the jet fuel (Kärcher et al., 2007; Lee et al., 2021), while $SO_4$ forms indirectly from the emissions of sulfur dioxide ($SO_2$) and sulfuric acid ($H_2SO_4$). Experimental studies indicate that aircraft convert approximately 97% of the sulfur content in fuel into $SO_2$, with $SO_2$ emissions nearly directly proportional to fuel sulfur levels. The remaining 2-3% is emitted as $H_2SO_4$ (Petzold et al., 2005; Jurkat et al., 2011; Owen et al., 2022). The gaseous conversion of $SO_2$ to $H_2SO_4$ involves a series of reactions (R1 – R3), where the oxidation of $SO_2$ by hydroxyl (OH) radicals ultimately results in the production of $H_2SO_4$ (Mikkonen et al., 2011):

$$SO_2 + OH \rightarrow HSO_3 \tag{R1}$$



$$HSO_3 + O_2 \rightarrow SO_3 + HO_2 \tag{R2}$$

$$SO_3 + 2H_2O \rightarrow H_2SO_4 + H_2O. \tag{R3}$$

The formation of SO₄ aerosols can proceed through multiple mechanisms. One pathway involves the condensation of $H_2SO_4$ vapor onto pre-existing particles, followed by particle growth through coagulation and further condensation (Whitby and McMurry, 1997; Laaksonen et al., 2000; Aquila et al., 2011). Sulfate aerosols may also form via the binary homogeneous nucleation between $H_2SO_4$ and $H_2O$, a process influenced by factors such as atmospheric relative humidity and temperature (Vehkamäki et al., 2002; Kaiser et al., 2014). Additionally, SO₄ production can occur through the reaction of $H_2SO_4$ with

gaseous ammonia ($NH_3$), resulting in the formation of ammonium sulfate (($NH_4)_2SO_4$) or ammonium bisulfate ($NH_4HSO_4$), which subsequently dissociate to yield SO₄, ammonium ($NH_4$) and hydrogen (Khoder, 2002). An alternative pathway involves the liquid-phase oxidation of sulfurous acid ($H_2SO_3$) by hydrogen peroxide ($H_2O_2$) and ozone ($O_3$; Martin and Damschen, 1981; Yang et al., 2017). Figure 1 illustrates the primary reaction mechanisms leading to SO₄ formation via $SO_2$ and $H_2SO_4$.

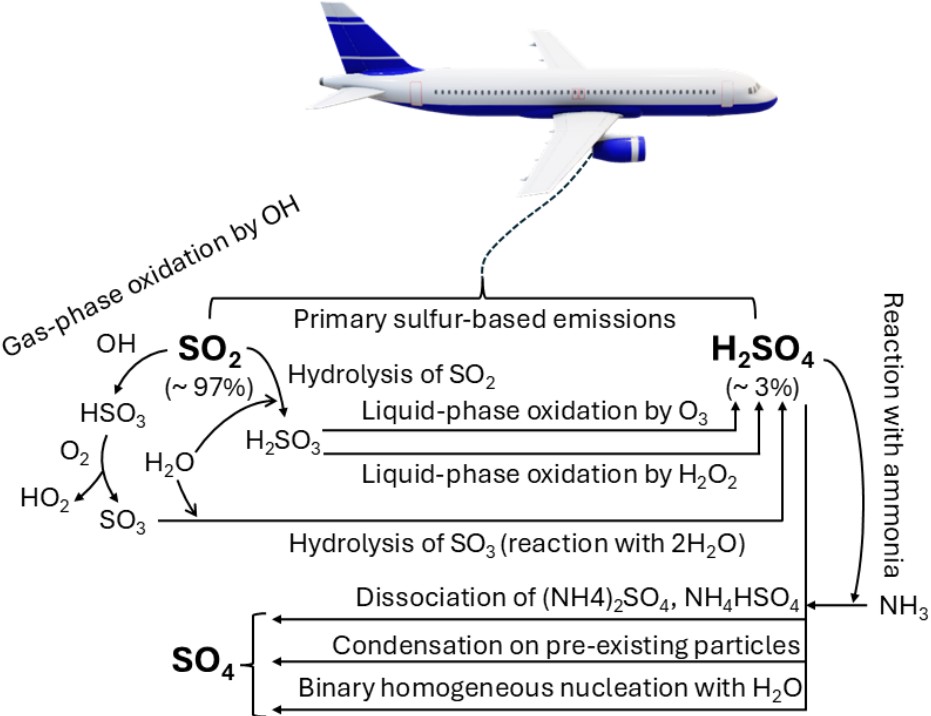

Figure 1 – Production mechanisms of sulfate (SO₄) from aviation emissions of sulfur dioxide ($SO_2$) and sulfuric acid ($H_2SO_4$).

Aerosol direct effects alter Earth's energy balance by scattering and absorbing solar and thermal infrared radiation depending on a particle's optical properties. Sulfate aerosols primarily scatter incoming shortwave solar radiation, resulting in a cooling





effect, whereas soot particles predominantly absorb this radiation, leading to a warming effect (Haywood and Shine, 1995; Kirkevåg et al., 1999; Lee et al., 2021). In addition to these direct effects, aerosols also have indirect effects through their influence on cloud properties, such as albedo and lifetime. These effects are known as the Twomey and Albrecht effects, respectively. As aerosols can often act as cloud condensation nuclei (CCN), an increase in their number can also increase the cloud droplet number concentration (CDNC) within liquid-phase clouds. Assuming the liquid water content within a cloud is constant, adding more CCNs will lead to more numerous cloud droplets, but with smaller radii. Given that cloud droplets mainly scatter solar radiation, an increase in CCNs therefore results in an overall enhancement in a cloud's reflective ability or albedo, an effect referred to as the Twomey effect (Twomey, 1977; Lohmann and Feichter, 1997; Sausen et al., 2012). Another consequence of having additional, smaller cloud droplets is the decrease of a cloud's precipitating ability, which prolongs its lifetime and therefore also the amount of solar radiation that it can reflect back into space (Albrecht, 1989; Lohmann and Feichter, 2005).

The considerably larger uncertainties in estimating indirect $SO_4$ effects compared to direct effects stem, among others, from the challenge in accurately describing the complex cloud microphysical processes occurring in the sub-grid scale that often need to be parameterized in global climate models (Lohmann and Feichter, 1997; Lohmann and Feichter, 2005). Additionally, limited understanding of aerosol transport pathways, including vertical transport and large-scale horizontal advection, as well as of the variations in microphysical and removal processes along these paths contribute to this uncertainty (Barrie et al., 2001; Weinzierl et al., 2017). These transport dynamics are particularly relevant in the context of aviation emissions, which occur at higher altitudes than ground-level sources such as shipping, as the emission location significantly influences both the production efficiency and atmospheric lifetime of pollutants, as has been shown for $NO_x$ emissions by Maruhashi et al. (2022). Model intercomparison studies, such as the Comparison of Large Scale Sulfate Aerosol Models (COSAM), have demonstrated that the conversion of $SO_2$ to $SO_4$ is challenging to predict, resulting in a spread of model results of $10 - 50\%$ and discrepancies of up to a factor of two compared to observational data from aircraft campaigns. This spread lead to uncertainties not only in the vertical distribution profiles of $SO_4$, but also in the resulting number of CCN produced, both of which directly influence estimates of $SO_4$ indirect forcing (Barrie et al., 2001; Penner et al., 2001). Such knowledge gaps therefore prevent a comprehensive assessment of aviation's net climate impact (Lee et al., 2021).

Haywood and Boucher (2000) provide estimates for anthropogenic $SO_4$ direct and indirect effects, with the latter spanning from -30 mWm$^{-2}$ to -178 mWm$^{-2}$. However, few studies have isolated these effects for aviation. Righi et al. (2013) assessed both, direct and indirect $SO_4$ effects, across the transport sector (land, shipping and aviation) and estimated aviation's contribution to lie within the range -69.5 mWm$^{-2}$ to 2.4 mWm$^{-2}$, noting that larger $SO_4$ particle sizes led to weaker cooling. Similarly, Gettelman and Chen (2013) found that smaller emitted particles (8 nm diameter) produced the strongest cooling effect at -160 mW·m$^{-2}$. Most existing studies assume instantaneous dilution and neglect plume-scale processes, likely overestimating particle number concentrations by at least 15% (Sharma et al., 2025).



Kapadia et al. (2016) demonstrated a linear relationship between aircraft fuel sulfur content (FSC) and the magnitude of the
Twomey effect, reporting RF values from -82.1 mWm$^{-2}$ (6000 ppm FSC) to -16.6 mWm$^{-2}$ (sulfur-free fuel). They also highlighted the policy trade-off between reduced cooling and improved air quality from low-sulfur fuels (see also Fuglestvedt et al., 2009).

Matthes et al. (2021) extended this analysis by examining the sensitivity of aviation-induced aerosol RF to cruise altitude, estimating a forcing of -15.1 mW·m$^{-2}$ at a reference altitude using the same simulation setup as Righi et al. (2013), but with
the REACT4C emissions inventory (Søvde et al., 2014) for 2006. They found that the cooling effect increased at lower altitudes. More recently, Righi et al. (2023) updated these estimates using the CMIP6 emissions inventory (Feng et al., 2020) for 2015 and calculated aviation's combined $SO_4$ direct and indirect effect to be -64 mWm$^{-2}$. Under the shared socioeconomic pathway (SSP) SSP2-4.5 scenario (Fricko et al., 2017) for 2050, this value increased to -126 mWm$^{-2}$. Table 1 summarizes these past findings on aviation-related $SO_4$ aerosol effects.

Table 1: An overview of studies that have estimated RF from aviation-induced $SO_4$ direct and indirect effects. The * indicates the amount of emitted $SO_2$ as referred in the comprehensive study of Lee et al. (2021). **Emitted $SO_2$ in Tg estimated assuming an emission index of 0.8 g/kg of fuel (Lee et al., 2010). FSC denotes fuel sulfur content.

| Study | Emission details | Calculation method | RF estimate [mWm$^{-2}$] |
|---|---|---|---|
| Righi et al., 2013 | CMIP5 emissions (Lamarque et al., (2010)) for the year 2000. Emission varies from 0.0008 to 0.1216 Tg($SO_2$)*. | Perturbation approach | Includes direct and indirect effects: RF = [-69.5, 2.4] |
| Gettelman and Chen, 2013 | Emission of 0.2257 Tg($SO_2$)* for the year 2006. | Perturbation approach | Sensitivity of indirect effect to $SO_4$ particle diameters: RF (8 nm) = -160 RF (14 nm) = -44 RF (21 nm) = -22 |
| Kapadia et al., 2016 | Range of emission cases for the year 2000 (FSC levels): Large = 2.358 Tg($SO_2$) Normal = 0.236 Tg($SO_2$) Low = 0.006 Tg($SO_2$) No sulfur = 0 Tg($SO_2$). | Perturbation approach | Only Twomey effect: $RF_{Large\ FSC}$ = -82.1 $RF_{Normal\ FSC}$ = -23.6 $RF_{Low\ FSC}$ = -16.8 $RF_{No\ FSC}$ = -16.6 |
| Matthes et al., 2021 | Simulation setup identical to Righi et al. (2013), but with the REACT4C emissions inventory for 2006 (Søvde et al., 2014). Emission ≈0.284 Tg($SO_2$)**. | Perturbation approach | Includes direct and indirect effects: -15.1 (reference case of aircraft flying at their optimal altitude) |
| Righi et al., 2023 | CMIP6 emissions (Feng et al., 2020) for 2015 and other SSP scenarios (e.g. SSP2-4.5). | Perturbation approach | Includes direct and indirect effects: $RF_{2015}$ = -64 $RF_{SSP2-4.5\ 2050}$ = -126 |

Within the context of climate-chemistry modeling, aviation's climate impacts are generally estimated using one of two main approaches: source apportionment or perturbation. The former is also referred to as tagging, as it involves labelling chemical



species and reactions of interest and accompanying their fate throughout a simulation (Wang et al., 2009). It is normally applied to quantify the contribution of a sector to the total concentration of a pollutant. The perturbation method, on the other hand, involves evaluating the marginal impact of a change in emissions typically by subtracting two simulations: one with all emissions and another with changed emissions (Blanchard, 1999; Hoor et al., 2009; Clappier et al., 2017). Based on Table 1 and to the authors' best knowledge, aviation-specific indirect effects of $SO_4$ have thus far only been estimated using the

perturbation method. While useful, it is insufficient on its own for formulating robust mitigation policies. Tagging techniques enable precise attribution of emissions to specific sectors, helping to identify those with the highest mitigation potential. The perturbation method can then complement this by quantifying the effects of targeted reduction measures (Mertens et al., 2018). Righi et al. (2023) highlight the difficulty of applying a tagging method for aerosols due to the challenges of tracking their complex liquid- and gas-phase transformations and interactions with clouds.

Only a handful of Eulerian and Lagrangian sulfate tagging schemes exist. Eulerian model studies include those by Yang et al. (2017), who used the Community Atmosphere Model (CAM) within the Community Earth System Model (CESM) framework to quantify global $SO_4$ contributions from 16 sources and assess both direct and indirect radiative forcing effects, between 2010 – 2014. Other Eulerian approaches target regional impacts in China. For example, Wu et al. (2017) utilized an online tagging method within the Nested Air Quality Prediction Model System (NAQPMS) to reveal that $SO_4$ levels in Shanghai

were significantly influenced by non-local sources. Similarly, Itahashi et al. (2017), employing the Particulate Source Apportionment Technology (PSAT) algorithm (Wagstrom et al., 2008) with the sixth version of the Comprehensive Air quality Model (CAMx), assessed $SO_4$ contributions from 31 Chinese provinces and found that emissions from the central and northern regions notably affected sulfate levels in Taiwan, Korea and Japan. Lagrangian approaches include the study by Riccio et al. (2014), which used the Hybrid Single-Particle Lagrangian Integrated Trajectory (HYSPLIT) model to show that $PM_{2.5}$ levels

in Naples were substantially impacted by emissions from other parts of Europe and dust transport from the Sahara. Despite their methodological differences, these studies highlight the influence of cross-territorial emissions on regional $SO_4$ levels as a result of long-range transport. There are also Lagrangian particle dispersion models like FLEXible PARTicle (FLEXPART) and LAGRANTO that have been applied to study sulfate transport and lifetime of $SO_2$ from volcanic emissions (Sun et al., 2023; Sun et al., 2024; Toohey et al., 2025). These studies are somewhat comparable to the scenario of aircraft cruise emissions,

as both emit $SO_2$ above sea level and may inject $SO_4$ into the upper troposphere and lower stratosphere. None of these studies, however, have characterized the aviation-specific global transport patterns of $SO_4$ originating from the primary emissions of $SO_2$ and $H_2SO_4$ introduced at subsonic cruise altitudes.

Here, we address this fundamental knowledge gap by introducing the first Lagrangian aerosol tagging submodel for $SO_4$ within the fifth-generation European Centre for Medium-Range Weather Forecasts – Hamburg (ECHAM)/Modular Earth Submodel

System (MESSy) Atmospheric Chemistry (EMAC) modeling framework, enabling a detailed, parcel-by-parcel analysis of aviation $SO_4$ transport patterns. AIRTRAC v2.0 represents an important advancement, extending the original Lagrangian



AIRTRAC submodel (Supplement of Grewe et al., 2014a) by incorporating aerosol processes along air parcel trajectories – capabilities unique to this updated version. Specifically, aerosol mixing ratios are computed using the extensively validated third-generation Modal Aerosol Dynamics model for Europe (MADE3; Kaiser et al., 2014; Kaiser et al., 2019) submodel, while aviation-specific contributions are quantified by AIRTRAC. Previously, AIRTRAC was limited to the study of gas-phase emissions such as $NO_x$ and $H_2O$ (Frömming et al., 2021; Maruhashi et al., 2022), primarily aimed at quantifying their contributions to atmospheric concentrations of reactive nitrogen species ($NO_y$), including nitric acid ($HNO_3$), $O_3$, the hydroperoxyl radical ($HO_2$), hydroxide (OH) and methane ($CH_4$). This Lagrangian approach offers a substantial computational advantage over more resource-intensive Eulerian simulations by enabling the simultaneous analysis of multiple emission scenarios (Maruhashi et al., 2024), while also providing clearer insight into the transport patterns of emitted pollutants (Frömming et al., 2021; Maruhashi et al., 2022).

In the present assessment, we focus on the $SO_4$ aerosol as it has been shown to be a highly efficient CCN for liquid clouds. In contrast, soot is hydrophobic and therefore generally becomes a CCN only when mixed internally with other more hygroscopic aerosols like $SO_4$ (Kristjánsson, 2002; Lee et al., 2021). This study has three primary objectives. First, it describes a novel Lagrangian aerosol-tagging submodel and the assumptions underlying its formulation. Second, the study demonstrates the usefulness of the new AIRTRAC v2.0 submodel in improving our understanding of the transport patterns of aviation-induced $SO_2$ and $H_2SO_4$ emissions, particularly in identifying where these lead to the largest $SO_4$ enhancements—especially over regions with abundant low-level liquid clouds. The analysis considers 28 globally distributed emission points along major present-day flight routes at a typical cruise altitude (240 hPa ≈ 10.6 km) for both winter and summer seasons. Using ESA satellite data to locate regions of significant liquid cloud cover, AIRTRAC v2.0 is applied to trace trajectories most likely to interact with these clouds, highlighting its potential to inform assessments of aerosol-cloud interactions. Third, the study compares the spatial distributions of $SO_4$ generated by this tagging approach against those derived using a perturbation method using a similar simulation setup. As part of our evaluation, we compare our $SO_2$ and $SO_4$ lifetime estimates with those reported in other Lagrangian modeling and observational studies of volcanic eruptions. The paper is structured into six sections: Section 1 provides an introduction; Section 2 describes the overall EMAC modeling setup, including the MADE3 aerosol submodel with which AIRTRAC v2.0 has been coupled within the MESSy framework, and presents the general formulation of the $SO_4$ mass transport equations. Section 3 outlines the AIRTRAC v2.0 infrastructure and details the tagging formulation of the $SO_4$ transport equations. Section 4 presents results of simulations from AIRTRAC v2.0 and integrates satellite cloud data to illustrate the submodel's capability to predict interactions with low-level liquid clouds. Section 5 compares AIRTRAC's output with the results of a perturbation approach and with prior studies. Finally, Section 6 summarizes key findings, considers limitations of the AIRTRAC v2.0 submodel and offers directions for future research.



## 2 Modeling framework

This section summarizes the most relevant characteristics of EMAC along with the main submodels applied in our base modeling setup (Section 2.1). Section 2.2 describes the $SO_2$ and $H_2SO_4$ pulse emission locations and inventory used in our
simulations. In Section 2.3, the submodel responsible for computing aerosol microphysics (MADE3), the most relevant aerosol microphysical processes and the general formulation of the $SO_4$ mass transport equations are presented. The submodels governing the removal processes and transport of aerosol species are then presented in Section 2.4.

### 2.1 The EMAC model setup

Chemistry-climate simulations in this assessment are performed with the EMAC model. It is a flexible, global model that
simulates a plethora of atmospheric processes along with interactions between land, ocean and anthropogenic activity by means of a coupling interface called MESSy (Jöckel et al., 2010) that can connect more than 100 different submodels to ECHAM5, EMAC's base general circulation model (Roeckner et al., 2006). Apart from AIRTRAC, which is responsible for calculating the contribution of aviation-related pollutants along air parcel trajectories and MADE3, which calculates aerosol dynamics and microphysical processes, there are other submodels that are also relevant in our modeling setup. Some of these are described
in Section 2.4. The full list of applied submodels is included in Table A1 in Appendix A.

Simulations are performed with version 5.3.02 of the base model ECHAM5 and with version 2.55.2 of the MESSy integrated framework. The EMAC model resolution is T42L41DLR, which corresponds to a quadratic Gaussian grid of size ~2.8º × 2.8º (with 128 longitude and 64 latitude grid cells) with 41 discrete, vertical hybrid sigma-pressure levels ranging from the surface to the uppermost layer of the atmosphere centered at 5 hPa. For comparability purposes with Righi et al. (2023), our model
output is characterized by a temporal resolution of 11 h with a model calculation time-step length of 15 min. The meteorology (the temperature, the wind divergence, the vorticity and the logarithm of the sea-level pressure) in our simulations has been nudged by Newtonian relaxation towards ERA-Interim reanalysis data (Dee et al., 2011) for the simulated year. Sea surface temperature and sea ice concentration have been prescribed from the ERA-Interim reanalysis data as well. Two simulations (starting on January 1, 2015 for Northern winter and on July 1, 2015 for Northern summer) were performed to accompany the
chemical transport of aviation $SO_2$ and $H_2SO_4$ and their conversion into $SO_4$ for three months across 28 emission points (see Fig. 2) at an altitude of 240 hPa. To ensure background meteorological conditions are in quasi-equilibrium, each simulation was preceded by a four-month spin-up period. Each emission point releases varying amounts of $SO_2$ and $H_2SO_4$ in the form of 15-minute pulse emissions. These are initially advected by 50 air parcels originating from the grid box of each emission point, consistent with the recommendation of Grewe et al. (2014a). These 50 air parcels are pseudo-randomly initialized around the
coordinates of an emission point according to a uniform distribution between 0 and 1. A simplified background chemistry mechanism is applied with the MECCA submodel (Sander et al., 2019) for the troposphere that involves the most relevant gaseous species like $NO_x$, $HO_x$, $CH_4$ and $O_3$. Aqueous-phase chemistry is handled by the SCAV submodel (Tost et al., 2006). These chemistry mechanisms are solved automatically by the Kinetic Pre-Processor (KPP) software using Fortran 90 code





(Sander et al., 2005). The overall Lagrangian setup is similar to the one used by Maruhashi et al. (2022) and Maruhashi et al.
(2024), while the applied aerosol chemistry and scavenging mechanisms are the same as those applied by Righi et al. (2023).

## 2.2 SO₂ and H₂SO₄ emission points

The 28 emission locations at which $SO_2$ and $H_2SO_4$ pulses are introduced (Fig. 2) are selected to reflect a realistic modern-day spatial distribution of aviation emissions. This distribution was determined according to the 2015 CMIP6 aviation emissions inventory that includes the correction for the latitudinal bias found by Thor et al. (2023). The exact coordinates of the 28 points
in Fig. 2 are included in Table B1 from Appendix B. Each coordinate has three dimensions – latitude, longitude and altitude – which are found by identifying the locations at which the aviation $SO_2$ mass flux (kg m⁻³ s⁻¹) is maximum. The emission altitude is determined by establishing the pressure level at which the zonally averaged aviation $SO_2$ mass flux is the largest (see Fig. B1 in Appendix B): ~240 hPa. We have developed an accompanying tool in Python (see EP_selector in Appendix B and the data repository of Maruhashi et al. (2025a) for the complete code) that approximates continuous emissions as a set of
distributed pulse emissions by automatically identifying the top 28 points whose grid cells have the largest $SO_2$ mass flux contributions (defined as a grid cell's area-weighted mean mass flux, see Eq. B1) within a user-defined mesh. The total mass of emitted $SO_2$ and $H_2SO_4$ across these points was scaled to yield the approximate global aviation $SO_2$ produced in a day from aviation in 2015 (Righi et al., 2023). The emitted amount of $H_2SO_4$ at any given emission point is determined by assuming that it constitutes 2% of the total $SO_2$ mass emitted by aviation, which is consistent with measurements of aircraft exhaust
plumes at cruise altitudes (Jurkat et al., 2011). Based on this assumption, the amount of $H_2SO_4$ emitted per simulation is found according to Eq. B3 in Appendix B.

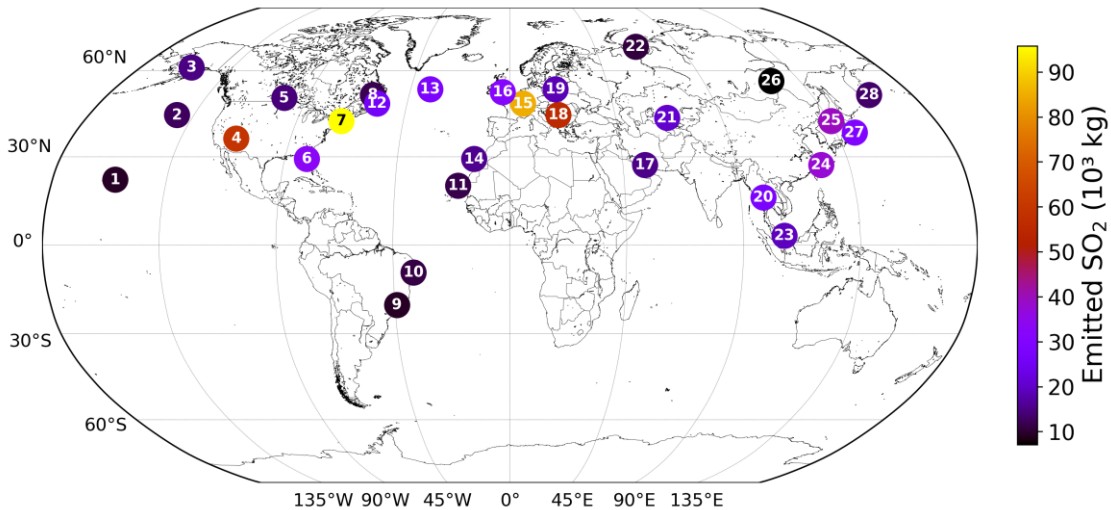

Figure 2 – The 28 emission points at which $SO_2$ and $H_2SO_4$ are emitted are represented as numbered circles. This horizontal distribution of points is shown for a pressure altitude of 240 hPa (~10.6 km). The amounts emitted at each point vary according





to the CMIP6 aviation emissions inventory and are distinguished by the color bar. The exact coordinates and emitted amounts
for $SO_2$ and $H_2SO_4$ are specified in Table B1 of Appendix B.

**2.3 The MADE3 submodel and the $SO_4$ mass transport equations**

Aerosol microphysical processes and dynamics are simulated using the MADE3 (Kaiser et al., 2014 and 2019) submodel, a
successor of the MADE (Ackermann et al., 1998; Lauer et al., 2005) and MADE-in (Aquila et al., 2011) submodels. The
performance of MADE3 has been extensively evaluated. Its ground-level aerosol mass concentrations have been compared to
data from a network of measurement stations and satellite observations. Its simulated vertical profiles of the mass mixing ratios
and particle number distributions have been evaluated against aircraft campaign data. Overall, the model has demonstrated
satisfactory alignment with observational data, although MADE3 tends to produce larger average sulfate concentrations near
the surface, with biases ranging from 13% to 92% when compared to observations from measurement stations (Kaiser et al.,
2019).

The MADE3 submodel considers nine types of aerosols: $SO_4$, BC, sea spray (Na), $H_2O$, chloride (Cl), mineral dust (DU), $NH_4$,
$NO_3$ and particulate organic matter (POM). Each aerosol type is further classified into nine modes, which result from the
combination of three size categories (Aitken (subscript 'k'), accumulation (subscript 'a') and coarse (subscript 'c')) and three
mixing states (soluble (subscript 's'), insoluble (subscript 'i') and mixed (subscript 'm')). MADE3 therefore adopts a modal
approach, where the total particle number distribution $n(\ln D)$ of an aerosol is obtained by the superposition of its nine
lognormal distributions (one for each mode M) according to Eq. 1a (Aquila et al., 2011; Kaiser et al., 2014; Boucher, 2015):

$$n(\ln D) = \sum_{M=1}^{9} \frac{N_M}{\sqrt{2\pi} \ln \sigma_M} \exp\left[-\frac{\left(\ln D - \ln D_{g,M}\right)^2}{2\ln^2 \sigma_{g,M}}\right], \qquad \text{(Eq. 1a)}$$

where $N_M$ is the total number concentration for mode M, $D_{g,M}$ is the median diameter and $\sigma_{g,M}$ is the geometric standard
deviation. Assuming spherical particles, the mass distribution $m(\ln D)$ is obtained by multiplying Eq. 1a by the particle density
$\rho_M$ and cubic diameter $D^3$ according to Eq. 1b:

$$m(\ln D) = \sum_{M=1}^{9} \frac{\pi}{6} \rho_M D^3 \frac{N_M}{\sqrt{2\pi} \ln \sigma_M} \exp\left[-\frac{\left(\ln D - \ln D_{g,M}\right)^2}{2\ln^2 \sigma_{g,M}}\right]. \qquad \text{(Eq. 1b)}$$

The Aitken mode is the smallest size range resolved by MADE3 and consists of particles around 10 nm, the accumulation
mode includes aerosols roughly 100 nm in diameter, and the coarse mode is the largest, with particle sizes of ~1 μm (Kaiser
et al., 2014). Figure 3 describes the differences across the three mixing states for $SO_4$. The soluble state (Fig. 3a) consists
entirely of soluble species like $SO_4$, $NO_3$ or others. Externally mixed or insoluble (Fig. 3b) aerosols contain a non-volatile core



(e.g. black carbon or mineral dust) with a total soluble mass fraction $\theta$ below 10% while internally mixed or mixed (Fig. 3c)
aerosols have the same structure, but $\theta$ is larger than 10%.

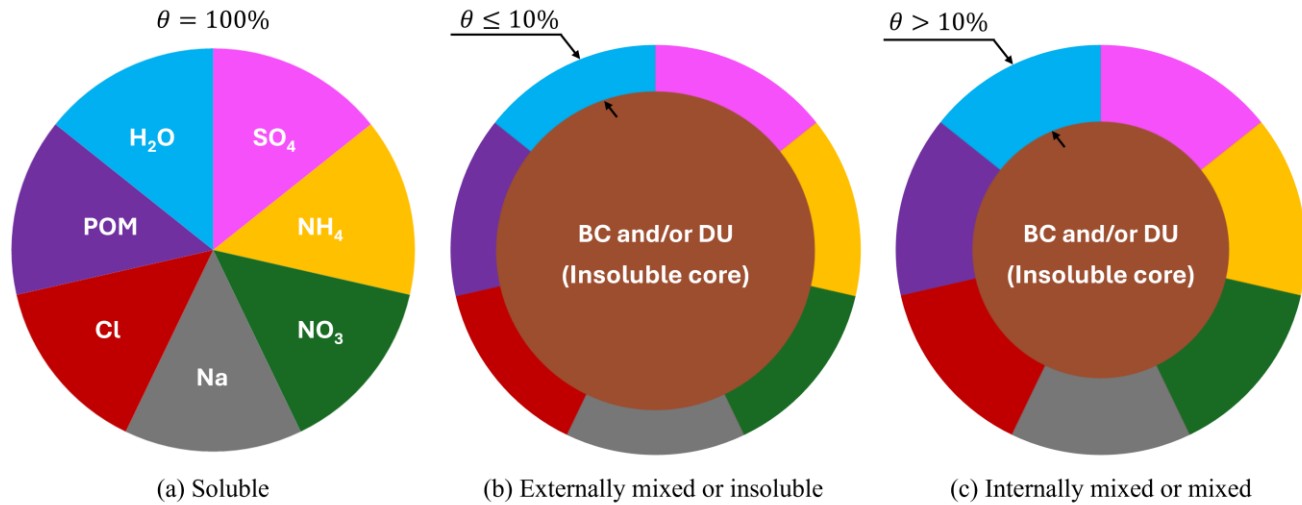

Figure 3 – The three aerosol mixing states according to MADE3. (a) Soluble state, where each of the seven colors represents
a soluble species, (b) externally mixed or "insoluble" state where the total soluble mass fraction $\theta$ is at most 10% and (c)
internally mixed or simply "mixed" state where the total soluble mass fraction $\theta$ is above 10%. BC and DU denote the insoluble
black carbon or mineral dust core, respectively. Figure is inspired by Kaiser et al. (2014).

The MADE3 submodel calculates tracer tendencies for both mass and particle number mixing ratios across these nine modes
and nine species. These tendencies encompass several microphysical processes: gas-particle partitioning (subscript "gtp"),
condensation (subscript "cond"), nucleation (subscript "nucl"), coagulation (subscript "coag"), particle growth (subscript "gr")
and aging (subscript "ag"). The general form of the governing differential equations for the mass mixing ratios $C_{I,M}$ of an
aerosol species I in mode M is therefore given by Eq. 2 (Aquila et al., 2011):

$$\frac{\partial C_{I,M}}{\partial t} = R(C_{I,M}) + \frac{\partial C_{I,M}}{\partial t}\bigg|_{gtp} + \frac{\partial C_{I,M}}{\partial t}\bigg|_{cond} + \frac{\partial C_{I,M}}{\partial t}\bigg|_{nucl} + \frac{\partial C_{I,M}}{\partial t}\bigg|_{coag} + \frac{\partial C_{I,M}}{\partial t}\bigg|_{gr} + \frac{\partial C_{I,M}}{\partial t}\bigg|_{ag}. \qquad \text{(Eq. 2)}$$

The term $R(C_{I,M})$ indicates the variation of aerosol mass concentrations as a result of removal (e.g. dry and wet deposition)
and transport phenomena like convection, advection or turbulent mixing processes that are all handled outside of MADE3 by
other submodels. These include sedimentation (handled by the SEDI submodel), dry deposition (handled by the DDEP
submodel), wet deposition (handled by the SCAV submodel) as well as mixing from atmospheric turbulence (handled in
Eulerian representation by the submodel E5VDIFF (Supplement of Emmerichs et al., 2021) and in Lagrangian representation
by the submodel LGTMIX (Brinkop and Jöckel, 2019)).




The focus of this study is on aerosols containing $SO_4$ ($I = SO_4$) and their nine modes, for which the following simplifications may be applied to Eq. 2:

1.  $\left.\dfrac{\partial C_{SO_4,M}}{\partial t}\right|_{gtp} = 0$. The gas-particle partitioning term is neglected for $SO_4$, as it is assumed that $H_2SO_4$ has an equilibrium vapor pressure that is low enough so that it is fully transferred from the gas to the aerosol phase at each time step. The reverse conversion is also not possible, i.e., $H_2SO_4$ cannot re-evaporate (Kaiser et al., 2014). For this reason, this process in MADE3 is handled by the coagulation routine and hence accounted for in the "coag" term.

       2.  $\left.\dfrac{\partial C_{SO_4,M\neq ks}}{\partial t}\right|_{nucl} = 0$. The nucleation term for $SO_4$ applies exclusively to the mass mixing ratio of the soluble Aitken
mode ($M = ks$) as freshly formed particles via the binary homogeneous nucleation between $H_2SO_4$ and $H_2O$ are assumed to instantaneously produce soluble $SO_4$ aerosols in the Aitken mode size range (Aquila et al., 2011; Kaiser et al., 2014; Kaiser et al., 2019).

       3.  $\left.\dfrac{\partial C_{SO_4,M=\{ks,as,cs\}}}{\partial t}\right|_{ag} = 0$. The aging process changes aerosols from the insoluble to the mixed modes when their soluble mass fractions exceed a critical threshold of 10%. Experimental studies (Weingartner et al., 1997; Khalizov et al.,
2009) have shown that aerosols with soluble mass fractions above this threshold will exhibit hygroscopic growth and expand by $H_2O$ uptake (Aquila et al., 2011). Consequently, the aging tendency only influences the insoluble and mixed modes.

By applying these three assumptions to Eq. 2, we derive the nine differential equations for the mass mixing ratio of each $SO_4$ aerosol mode (Eq. 3). Each term is explained in greater detail in the coming sections.

$$
\quad \frac{\partial C_{SO_4,M}}{\partial t} = \begin{cases} R\left(C_{SO_4,M}\right) + \left.\dfrac{\partial C_{SO_4,M}}{\partial t}\right|_{cond} + \left.\dfrac{\partial C_{SO_4,M}}{\partial t}\right|_{nucl} + \left.\dfrac{\partial C_{SO_4,M}}{\partial t}\right|_{coag} + \left.\dfrac{\partial C_{SO_4,M}}{\partial t}\right|_{gr} & ; M \equiv ks \\ R\left(C_{SO_4,M}\right) + \left.\dfrac{\partial C_{SO_4,M}}{\partial t}\right|_{cond} + \left.\dfrac{\partial C_{SO_4,M}}{\partial t}\right|_{coag} + \left.\dfrac{\partial C_{SO_4,M}}{\partial t}\right|_{gr} + \left.\dfrac{\partial C_{SO_4,M}}{\partial t}\right|_{ag} & ; M \equiv ki, km, ai, am, ci, cm \\ R\left(C_{SO_4,M}\right) + \left.\dfrac{\partial C_{SO_4,M}}{\partial t}\right|_{cond} + \left.\dfrac{\partial C_{SO_4,M}}{\partial t}\right|_{coag} + \left.\dfrac{\partial C_{SO_4,M}}{\partial t}\right|_{gr} & ; M \equiv as, cs \end{cases} \quad \text{(Eq. 3)}
$$

### 2.3.1 The nucleation tendency

Homogeneous nucleation between $H_2SO_4$ and $H_2O$ is a critical process behind the formation of $SO_4$ aerosols in the atmosphere and is calculated in MADE3 by means of a parameterization by Vehkamäki et al. (2002) for the nucleation rate J. The parameterization of J is applicable for atmospheric temperatures T between 230.15 K and 305.15 K (typical of the troposphere
and stratosphere), relative humidities (RH) from 0.01% and 100% and sulfuric acid concentrations ($C_{H_2SO_4}$) between $10^4$ and $10^{11}$ molecules per $cm^3$. In an earlier version of MADE3 (version 2.0b; Kaiser et al., 2014), it was assumed that the newly



nucleated sulfate particles were all characterized by a representative wet diameter of 3.5 nm, although in reality they are likely to be smaller and grow to more comparable sizes via other processes like coagulation and condensation, as is acknowledged by Binkowski and Roselle (2003). In the more recent MADE3 release (version 3.0; Kaiser et al., 2019), freshly formed particles

are considered to have a dry diameter of 10 nm to implicitly account for the rapid growth to larger sizes within a few hours, a phenomenon that cannot yet be resolved by global models. Compared to tropospheric observations of nucleated particles gathered by other studies (Modini et al., 2009; Ueda et al., 2016), such a modification appears to yield more accurate simulation results for the size distribution of aerosol particles and their particle number concentrations. The calculation of the $SO_4$ nucleation tendency follows Eq. 4:

$$\frac{\partial C_{SO_4,ks}}{\partial t}\bigg|_{nucl} = J\big(T, RH, C_{H_2SO_4}\big) \times m_{10nm}(RH) \times \exp\left[\frac{9}{2}\ln^2 \sigma_{ks}\right].  \qquad (Eq.\,4)$$

The term $m_{10nm}(RH)$ denotes the mass of a freshly nucleated spherical $SO_4$ particle with a dry diameter of 10 nm, which depends on the ambient RH. The factor $\exp\left[\frac{9}{2}\ln^2 \sigma_{ks}\right]$ relates to the geometric standard deviation of the soluble Aitken mode ($\sigma_{ks}$) for a lognormal distribution (see explanation of Eq. 1a). It is worth highlighting that nucleation involves only the amount of $H_2SO_4$ that has not yet been consumed by condensation. In reality, both processes simultaneously compete for $H_2SO_4$, but

within the MADE3 code, the condensation process has been set to occur before nucleation, being the slower one. This consequently amplifies the influence of $SO_4$ nucleation at locations where condensational sinks for $H_2SO_4$ are low (Kaiser et al., 2014).

**2.3.2 The condensation tendency**

The condensation of $H_2SO_4$ onto pre-existing particles leads to an overall gain in the mass concentration of $SO_4$. As was
mentioned, it is assumed that all of the sulfuric acid is converted entirely from the gas to the aerosol phase and that no re-evaporation occurs in the opposite direction (Kaiser et al., 2014). This $SO_4$ gain ($\frac{\partial C_{SO_4,M}}{\partial t}\big|_{cond}$) is quantified by Eq. 5 and may be expressed in terms of the dimensionless coefficients $\Omega_{SO_4,M}$ and the amount of condensed $H_2SO_4$, represented by $\Delta H_2SO_4|_{cond}$, for each aerosol mode M (Whitby et al., 1991; Aquila et al., 2011):

$$\frac{\partial C_{SO_4,M}}{\partial t}\bigg|_{cond} = \Omega_{SO_4,M} \times \frac{\Delta H_2SO_4|_{cond}}{\Delta t}.  \qquad (Eq.\,5)$$

The coefficients $\Omega_{SO_4,M}$ measure the modal contribution of the $SO_4$ growth term for mode M ($G^{(3)}_{SO_4,M}$), which is proportional to the time rate of change of the total volume of aerosol particles in a specific mode M, relative to the sum of all nine modal growth terms, as expressed by Eq. 5a. The variable $\Delta H_2SO_4|_{cond}$ is calculated by solving a first-order linear differential



equation (Eq. 5b) that describes the difference between the gas-phase production rate of sulfuric acid (P) and its condensational loss rate L (Aquila et al., 2011; Kaiser et al., 2014):


$$\Omega_{SO_4,M} = \frac{G^{(3)}_{SO_4,M}}{\sum_{j=1}^{9} G^{(3)}_{SO_4,j}},$$

(Eq. 5a)

$$\frac{dC_{H_2SO_4}(t)}{dt} = P - L \cdot C_{H_2SO_4}(t).$$

(Eq. 5b)

The growth rates G featured in Eq. 5a may be computed by finding the harmonic mean of the growth rates for the $SO_4$ aerosol in mode M for two regimes: the free-molecular regime in which collisions between gas molecules are not as frequent and the near-continuum regime in which collisions are frequent enough for the gas to be considered a continuous fluid. Other variables

like the saturation vapor pressure and diffusion coefficients of sulfuric acid are also necessary to calculate $\Omega_{SO_4,M}$. Further details on the calculation of the terms in Eq. 5 are provided by Aquila et al. (2011) and Kaiser et al. (2014).

### 2.3.3 The coagulation tendency

Two types of coagulation processes are distinguished: intra- and intermodal. The former occurs between particles of the same mode and produces aerosols that belong to the same mode as the original ones. The latter occurs between particles from

different size and mixing states and results in particles with diameters comparable to the larger of the colliding aerosols. Only intermodal coagulation is therefore relevant for the spatio-temporal evolution of aerosol mass mixing ratios. The contribution to mode M of $SO_4$ resulting from the coagulation of an aerosol in mode $p$ and another in mode $q$ is given by Eq. 6 (Kaiser et al., 2014):

$$\left.\frac{\partial C_{SO_4,M}}{\partial t}\right|_{coag} = \frac{\pi}{6}\sum_{p=1}^{9}\sum_{q=1}^{9}\left[\left(\delta_{M,\tau_{pq}} - \delta_{M,p}\right) \cdot \frac{C_{SO_4,p}}{\sum_{s=1}^{A} C_{s,p}} \cdot \rho_p \int_0^{\infty}\int_0^{\infty} (D_1)^3 \beta(D_1,D_2) n_p(D_1) n_q(D_2) dD_1 dD_2 + \cdots\right.$$

(Eq. 6)

$$\left.\cdots + \left(\delta_{M,\tau_{pq}} - \delta_{M,q}\right) \cdot \frac{C_{SO_4,q}}{\sum_{s=1}^{A} C_{s,q}} \cdot \rho_q \int_0^{\infty}\int_0^{\infty} (D_2)^3 \beta(D_1,D_2) n_p(D_1) n_q(D_2) dD_1 dD_2\right].$$

The resulting mode for each type of collision in MADE3 is governed by a categorical variable $\tau_{pq}$ where $p$ and $q$ are the

colliding modes. The possible values of $\tau_{pq}$ are detailed in Table 2 from Kaiser et al. (2014), having also a dependency on the particle's soluble mass fraction $\theta$ relative to water. The symbol $\delta_{x,y}$ denotes the Kronecker delta, which by definition, assumes a unit value only when both subscripts $x$ and $y$ refer to the same aerosol mode and being zero otherwise. The mass mixing ratios of $SO_4$ corresponding to modes $p$ and $q$ are indicated by $C_{SO_4,p}$ and $C_{SO_4,q}$ and their densities by $\rho_p$ and $\rho_q$ respectively.



The upper bound A of the summations indicates the total number of tracer species s in MADE3, i.e., SO₄, NH₄, NO₃, Na, Cl,

POM, BC, DU and H₂O. The particle number distribution for modes $p$ and $q$ with particle diameters $D_1$ and $D_2$, are written as $n_p(D_1)$ and $n_q(D_2)$, respectively. Lastly, the Brownian coagulation kernels for particles with diameters $D_1$ and $D_2$ are expressed as $\beta(D_1, D_2)$, which vary according to the flow regime, i.e., free-molecular or near-continuum. From a more intuitive perspective, they describe the probability of collision between particles of diameters $D_1$ and $D_2$ according to Brownian motion. Their complete mathematical formulations may be consulted from Aquila et al. (2011) and from Kaiser et al. (2014).

**2.3.4 The growth and aging tendency**

Although in theory the growth and aging processes are distinct, as in Eq. 2, they are coupled and therefore not given each their own tendency in MADE3. The collective tendency in MADE3 that describes growth and aging is called "rename" and is impacted by both, the condensation and the coagulation processes. The growth process, for instance, refers to the possibility of aerosols growing as they either coagulate or condense onto other pre-existing particles. This may lead to a redistribution in

the aerosol modes as particles grow and are reclassified from the Aitken to the accumulation mode. The aging process, in turn, relates to the transformation of insoluble particles via the acquisition of a soluble coating that alters their mixing state from hydrophobic to hydrophilic. By default, if the soluble mass fraction of an insoluble mode reaches a threshold of 10%, the mode is turned to a mixed mode. Aging therefore only impacts the insoluble and mixed modes of SO₄ (Aquila et al., 2011; Kaiser et al., 2014).

**2.4 The aerosol removal processes and transport**

The removal processes included in the transport term $R(C_{SO_4,M})$ of Eq. 3 are dry deposition, sedimentation and scavenging of aerosols. Dry deposition refers to the removal of atmospheric aerosols through various interactions with surfaces, such as land or sea, and includes mechanisms such as impaction, interception and diffusion (Farmer et al., 2021). Although sedimentation is a subset of dry deposition, there are two notable differences between them in the coding of each process.

Firstly, sedimentation is handled by the SEDI submodel (Kerkweg et al., 2006a) and applies to the entirety of the simulation's vertical domain, whereas dry deposition, handled by the DDEP submodel (Kerkweg et al., 2006a), only applies to the lowermost layer of the model. Secondly, sedimentation solely affects the removal of aerosols and not of gaseous species due to mass considerations.

The dry deposition flux for aerosols is proportional to the dry deposition velocity $v_d$, which can vary according to the surface

type: vegetation (subscript "veg"), soil and snow (subscript "slsn") and water (subscript "wat"). The dry deposition tendency for the SO₄ aerosol is represented according to Eq. 7, where parameters $\beta_1$, $\beta_2$ and $\beta_3$ depend on the surface type:



$$\frac{\partial C_{SO_4,M}}{\partial t}\Big|_{DDEP} \propto \underbrace{\beta_1 \times v_{d,\text{veg}}(SO_4) + \beta_2 \times v_{d,\text{slsn}}(SO_4) + \beta_3 \times v_{d,\text{wat}}(SO_4)}_{v_d}. \tag{Eq. 7}$$

The sedimentation tendency (Eq. 8) is in turn proportional to the terminal sedimentation velocity $v_t$, which depends on the Stokes velocity $v_{\text{Stokes}}$, the Cunningham slip flow correction factor ($f_{\text{csf}}$) and the Slinn factor ($f_s$). The latter is used to correct for the larger average sedimentation velocity of the lognormal population of aerosols when compared to the sedimentation velocity of a particle with an average particle diameter:

$$\frac{\partial C_{SO_4,M}}{\partial t}\Big|_{SEDI} \propto \underbrace{v_{\text{Stokes}}(M) \times f_{\text{csf}}(M) \times f_s(M)}_{v_t}. \tag{Eq. 8}$$

The expressions for the terms needed to compute both dry deposition and sedimentation velocities in Eqs. 7 and 8, are documented by Kerkweg et al. (2006a).

Wet deposition of $SO_4$ refers to scavenging as a result of the precipitation of either ice or liquid water. Aerosol lifetimes are therefore strongly influenced by the interaction between wet and dry deposition processes. Global studies have shown that wet deposition is more important for removing aerosols, but dry deposition is naturally dominant in cloud-free regions where precipitation is unlikely (Farmer et al., 2021). Wet deposition is calculated by the SCAV submodel (Tost et al., 2006) and considers two mechanisms: nucleation scavenging (rainout) and impaction scavenging (washout). The former involves the removal of chemical species from the atmosphere by means of nucleation and subsequent growth of cloud droplets that dissolve them and are then rained out. The latter process refers to the removal of aerosols and gases via their direct collision with raindrops. In a cloud-free region, nucleation scavenging is disregarded and only impaction scavenging is considered. Both mechanisms are contemplated via parameterizations that depend on, e.g., the Brownian motion of aerosols. These are described in further detail by Tost et al. (2006).

The transport of $SO_4$ in a Lagrangian framework is made possible primarily by three submodels: ATTILA, LGTMIX and TREXP. The Atmospheric Tracer Transport in a LAgrangian (ATTILA; Reithmeier and Sausen, 2002; Brinkop and Jöckel, 2019) submodel is the transport scheme for the Lagrangian air parcels and it resolves their advection according to the EMAC wind field and their convective motion. The LaGrangian Tracer MIXing (LGTMIX; Brinkop and Jöckel, 2019) submodel estimates the mass exchange of different chemical species from the isotropic mixing that occurs from atmospheric turbulence. Lastly, although the Tracer Release Experiments from Point sources (TREXP; Jöckel et al., 2010) submodel is not directly involved in the transport of tracers, it assists ATTILA and LGTMIX by defining the initial emission conditions (i.e. position and time of amount emitted) for point sources.



## 3 The new AIRTRAC v2.0 submodel

The AIRTRAC submodel (Supplement of Grewe et al., 2014a) was originally developed to improve our understanding of
some of aviation's gas-phase emissions, more specifically that of $NO_x$ and $H_2O$. The new AIRTRAC v2.0 submodel now has
expanded capabilities to calculate the contributions of aircraft $H_2SO_4$ and $SO_2$ emissions to the nine aerosol modes of $SO_4$. As
in the gas-phase scheme, source contributions are computed using the tagging approach of Grewe (2013), which is shown in
Eq. 9, and on the method applied by Itahashi et al. (2017) and Wu et al. (2017) (hereafter IW17). In IW17, $SO_4$ microphysical
processes (e.g. coagulation and condensation) are scaled according to precursor emissions such as $SO_2$, while removal
processes (dry and wet deposition) are scaled relative to the remaining aviation-attributable $SO_4$. State variables, like the
different aerosol modes of $SO_4$, are denoted by $x_M$, where x specifies the chemical species and M the mode. The index j refers
to the number of tagging categories that together add up to the quantity $x_M$. The term $P_M^j(t)$ indicates the time-dependent
contribution of tagging category j to mode M and $F_M$ is the state-dependent forcing for mode M, which could represent
production and/or loss terms associated with a state variable $x_M$.

$$\frac{\partial}{\partial t}x_M^j = P_M^j(t) + F_M(\mathbf{x})\frac{\mathbf{x}^{j^T}\nabla F_M(\mathbf{x})}{\mathbf{x}^T \cdot \nabla F_M(\mathbf{x})}$$
(Eq. 9)

This approach entails the calculation of a fractional weight, $\frac{\mathbf{x}^{j^T}\nabla F_M(\mathbf{x})}{\mathbf{x}^T \cdot \nabla F_M(\mathbf{x})}$, that scales the total forcing term $F_M(\mathbf{x})$ to isolate the
forcing component attributable to tagging category j for mode M. The numerator represents the weighted influence of tagging
category j on the total forcing $F_M$ while the denominator is the total forcing from all categories. This approach ensures that the
sum of the tagged contributions $x_M^j$ from all categories equals the solution of the original ordinary differential equation (ODE)
for the untagged total $x_M$ (Grewe, 2013). In the context of this study, two tagging categories are distinguished: aviation (index
"avi") and the remaining sources (index "rem"), which include anthropogenic activity, other transport sectors like shipping
and biogenic emissions. This fraction is applied to the MADE3 tendencies, allowing the model to calculate aviation-attributable
contributions to $SO_4$ enhancement. By construction, this guarantees a closed budget for the total amount of $SO_4$ in the
atmosphere, as the sum of the individual sources mathematically equates to the total (Eq. 10).

$$\frac{\partial C_{SO_4,M}}{\partial t} = \frac{\partial C_{SO_4,M}}{\partial t}\bigg|^{avi} + \frac{\partial C_{SO_4,M}}{\partial t}\bigg|^{rem}$$
(Eq. 10)

### 3.1 Submodel infrastructure

AIRTRAC v2.0 leverages the same technical infrastructure of its predecessor, introducing new Lagrangian tracers dedicated
to $SO_4$, $SO_2$ and $H_2SO_4$ chemistry. Some of these include tracers specific to individual aerosol microphysical processes such
as coagulation, enabling the estimation of each process's influence on an aerosol mode. A new subroutine called





"airtrac_aerosol_integrate" was created to handle the calculation of aerosol chemistry along the Lagrangian air parcel trajectories. The AIRTRAC control (CTRL) and coupling (CPL) namelists have also been adapted to now allow the user to select the value of "airtrac_mode", which should be set to "1" when studying $SO_4$ aerosols and to "2" for the gas-phase analysis mode.

## 3.2 Tracking aviation $SO_2$ and $H_2SO_4$ emissions

As $SO_2$ and $H_2SO_4$ are both precursors of $SO_4$ and are directly emitted by aircraft, their evolution throughout the course of a simulation must be tracked to properly account for aviation's impact on sulfate. The key gas-phase, tropospheric reaction involving $SO_2$ that leads to the production of $H_2SO_4$ in the chemistry mechanism adopted in this study is represented by the net Reaction R4:

$$SO_2 + OH \rightarrow H_2SO_4 + HO_2. \tag{R4}$$

The evolution of aviation $SO_2$ is simplified and described in AIRTRAC as a pure loss process, where it is oxidized to form $H_2SO_4$ according to Reaction R4. While processes such as scavenging contribute to $SO_2$ removal, these are excluded from the analysis due to the computational constraints arising from the difficulty of storing all of the liquid-phase tracers associated with the production of $SO_4$ from $SO_2$ (Tost et al., 2006). These removal processes primarily affect liquid-phase species anyway and thus have a smaller impact on the simulated gas-phase $SO_2$. Consequently, the mixing ratio of aviation-attributable $SO_2$

$(C_{SO_2}|^{avi})$, is based on the initial amount emitted into the atmosphere $(C_{SO_2}(t = 0))$ and the production rate of $H_2SO_4$ from $SO_2$ that is provided by the MECCA submodel $(P_{H_2SO_4})$, as is illustrated by Eq. 11. Unlike in Eq. 3, the term $R(C_{SO_2})$ only contemplates the effects of isotropic turbulent mixing:

$$C_{SO_2}\Big|^{avi} = \underbrace{C_{SO_2}(t = 0)}_{Emission} - \underbrace{\frac{C_{SO_2}\Big|^{avi}}{C_{SO_2}\Big|^{avi} + C_{SO_2}\Big|^{rem}}}_{Tagging\ ratio} \times \underbrace{\frac{M_{SO_2}}{M_{H_2SO_4}}}_{Molar\ masses} \times \underbrace{P_{H_2SO_4}}_{Production\ rate} \times \Delta t + \underbrace{R(C_{SO_2})}_{Turbulence}. \tag{Eq. 11}$$

In theory, the tagging ratio that scales the aviation-attributable $SO_2$ in Eq. 11 should consider the amount of OH emanating

from both, aviation and other sources, and therefore be expressed as $\frac{1}{2} \times \left( \frac{C_{SO_2}|^{avi}}{C_{SO_2}|^{avi} + C_{SO_2}|^{rem}} + \frac{C_{OH}|^{avi}}{C_{OH}|^{avi} + C_{OH}|^{rem}} \right)$, corresponding to the case of bimolecular reactions by Grewe et al. (2010) and Grewe (2013). However, this contribution is omitted in Eq. 11, as this study focuses exclusively on the influence of aviation $SO_2$ and $H_2SO_4$ emissions on $SO_4$. Since OH is not directly emitted by aircraft, fully tracking it would at least require other gas-phase aircraft emissions like $NO_x$ and CO and their chemical cycling in the atmosphere with other compounds like $HO_2$ and $O_3$ to be completely followed, which is beyond the



scope of this assessment. The production rate $P_{H_2SO_4}$ is the amount of $H_2SO_4$ produced from the oxidation of $SO_2$, so it must

be converted into the equivalent amount of $SO_2$ loss by multiplying it by the ratio of molar masses ($\frac{M_{SO_2}}{M_{H_2SO_4}}$).

To track the evolution of gas-phase, aviation-attributable $H_2SO_4$, the amount produced from the conversion of $SO_2$ described in Eq. 11 must be considered along with two primary sinks: the binary nucleation of $H_2SO_4$ with $H_2O$ and the condensation of $H_2SO_4$ onto pre-existing particles, as is represented in Eq. 12. As a limitation to the approach represented by Eq. 11 and Eq.

12, only the gas-phase formation of $H_2SO_4$ (produced from the hydrolysis of $SO_3$ via Reactions R1 – R3) is considered in our analysis. The liquid-phase production that involves the oxidation by $O_3$ and hydrogen peroxide ($H_2O_2$) of $H_2SO_3$, which in turn is formed by the hydrolysis of $SO_2$ (Sheng et al., 2018; Shostak et al., 2019), is excluded. While this pathway is the dominant source of sulfuric acid (Textor et al., 2006), incorporating it is challenging, due to the cloud evaporation assumption described by Tost et al. (2006), which considers that clouds and aqueous-phase species are fully evaporated at the end of each time step.

This assumption is used to avoid the high computation and memory costs that would otherwise be required to advect both gas- and aqueous-phase tracer species. The term $R(C_{H_2SO_4})$ therefore again only represents isotropic turbulent mixing.

$$C_{H_2SO_4}\big|^{avi} = \underbrace{C_{H_2SO_4}(t=0)}_{Emission} + \underbrace{\frac{C_{SO_2}\big|^{avi}}{C_{SO_2}\big|^{avi}+C_{SO_2}\big|^{rem}} \times P_{H_2SO_4} \times \Delta t}_{Oxidation\ of\ SO_2} + \cdots$$

$$\cdots - \underbrace{\frac{C_{H_2SO_4}\big|^{avi}}{C_{H_2SO_4}\big|^{avi}+C_{H_2SO_4}\big|^{rem}} \times \frac{\partial C_{H_2SO_4}}{\partial t}\bigg|^{avi}_{nucl} \times \Delta t}_{Nucleation\ of\ H_2SO_4} + \cdots \quad (Eq.\ 12)$$

$$\cdots - \underbrace{\frac{C_{H_2SO_4}\big|^{avi}}{C_{H_2SO_4}\big|^{avi}+C_{H_2SO_4}\big|^{rem}} \times \frac{\partial C_{H_2SO_4}}{\partial t}\bigg|^{avi}_{cond} \times \Delta t}_{Condensation\ of\ H_2SO_4} + \underbrace{R(C_{H_2SO_4})}_{Turbulence}$$

**3.3 Aviation's contribution to $SO_4$ via nucleation and condensation of $H_2SO_4$**

The contribution from aviation to the formation of the soluble Aitken mode of $SO_4$ via the binary nucleation of $H_2SO_4$-$H_2O$ is estimated by applying a tagging ratio to the MADE3 nucleation tendency (Eq. 4). This tagging ratio in Eq. 13 naturally involves

$H_2SO_4$, given that sulfuric acid from aviation ($C_{H_2SO_4}\big|^{avi}$) drives the nucleation process.

$$\frac{\partial C_{SO_4,ks}}{\partial t}\bigg|^{avi}_{nucl} = \frac{C_{H_2SO_4}\big|^{avi}}{C_{H_2SO_4}\big|^{avi}+C_{H_2SO_4}\big|^{rem}} \times \frac{\partial C_{SO_4,ks}}{\partial t}\bigg|_{nucl} \quad (Eq.\,13)$$



Aviation's contribution to $SO_4$ via the condensation of $H_2SO_4$ is estimated in a fashion similar to Eq. 13, as the tagging ratio is exactly the same (Eq. 14) due to the central role of $H_2SO_4$ also in this process. The main difference is that the condensation process may contribute to increasing any one of the nine $SO_4$ aerosol modes.


$$\frac{\partial C_{SO_4,M}}{\partial t}\Bigg|_{cond}^{avi} = \frac{C_{H_2SO_4}\big|^{avi}}{C_{H_2SO_4}\big|^{avi} + C_{H_2SO_4}\big|^{rem}} \times \frac{\partial C_{SO_4,M}}{\partial t}\Bigg|_{cond} \qquad \text{(Eq. 14)}$$

### 3.4 Aviation's contribution to $SO_4$ via particle coagulation, growth and aging

We first derive an expression for aviation's contribution to the soluble Aitken mode (M = ks) of $SO_4$ via coagulation by applying the tagging formulation in Eq. 9. For clarity and conciseness in representing the full coagulation equation (Eq. 6), vector notation is introduced for the state variables $\mathbf{x}$ according to Eq. 15 for all nine aerosol modes, where $x_1$, for instance, is

the mass concentration of mode "ks", i.e., $x_1 \equiv C_{SO_4,ks}$. Each state variable $x_i$ represents the sum of contributions from all sources, in this study the sources are $j = \{avi, rem\}$.

$$\mathbf{x} = \begin{bmatrix} x_1 \\ x_2 \\ x_3 \\ x_4 \\ x_5 \\ x_6 \\ x_7 \\ x_8 \\ x_9 \end{bmatrix} = \begin{bmatrix} x_1^{avi} + x_1^{rem} \\ x_2^{avi} + x_2^{rem} \\ x_3^{avi} + x_3^{rem} \\ x_4^{avi} + x_4^{rem} \\ x_5^{avi} + x_5^{rem} \\ x_6^{avi} + x_6^{rem} \\ x_7^{avi} + x_7^{rem} \\ x_8^{avi} + x_8^{rem} \\ x_9^{avi} + x_9^{rem} \end{bmatrix} = \begin{bmatrix} C_{SO_4,ks} \\ C_{SO_4,km} \\ C_{SO_4,ki} \\ C_{SO_4,as} \\ C_{SO_4,am} \\ C_{SO_4,ai} \\ C_{SO_4,cs} \\ C_{SO_4,cm} \\ C_{SO_4,ci} \end{bmatrix} \qquad \text{(Eq. 15)}$$

The coagulation kernels are also more compactly rewritten (Eqs. 16a and 16b), where $p$ and $q$ are the modes of the colliding particles. Additionally, the summation in the denominator of Eq. 6 for an aerosol in mode $p$ with a diameter $D_1$ will be

reformulated more succinctly as $\sum_{s=1}^{A=9} C_{s,p} = x_1 + C_p$, where $C_p = C_{NH_4,p} + C_{NO_3,p} + C_{Na,p} + C_{Cl,p} + C_{POM,p} + C_{BC,p} + C_{DU,p} + C_{H_2O,p}$. We also note that $f_{p,q} = f'_{q,p}$.

$$f_{p,q} = \int_0^\infty \int_0^\infty (D_1)^3 \beta(D_1, D_2) n_p(D_1) n_q(D_2) dD_1 dD_2 \qquad \text{(Eq. 16a)}$$

$$f'_{p,q} = \int_0^\infty \int_0^\infty (D_2)^3 \beta(D_1, D_2) n_p(D_1) n_q(D_2) dD_1 dD_2 \qquad \text{(Eq. 16b)}$$





When evaluating the double summations of Eq. 6 for the case M = ks, it is worth noting that for intermodal coagulation, i.e., $p \neq q$, the value of $\tau_{pq} \neq$ ks. In other words, when two particles from different modes collide, the destination mode will never be the soluble Aitken mode (Table 2 from Kaiser et al., 2014). It follows that since $\tau_{pq} \neq$ ks, the Kronecker deltas depending on $\tau_{pq}$ will be zero: $\delta_{ks,\tau_{pq}} = 0$. In contrast, the Kronecker deltas $\delta_{ks,p}$ and $\delta_{ks,q}$ evaluate to "1" whenever either mode $p$ or $q$ is also ks. We note that if $p = q$, the coagulation tendency is zero, as intramodal collisions do not affect mass concentrations. It is also essential to recall that the destination mode of the coagulated particle depends on the soluble mass fraction $\theta$ of the final particle. This dependency implies that the coagulation tendency for each of the nine modes will be a piecewise function ($F_1$) in terms of $\theta$. Combining Eqs. 6 and 16 leads to the more compact formulation of the coagulation tendency for mode ks (Eq. 17).

$$\frac{\partial x_1}{\partial t}\Big|_{\text{coag}} = F_1(\mathbf{x}) = \begin{cases} -A \dfrac{x_1}{x_1 + C_1}; & \theta = 1 \text{ (Soluble)} \\[2mm] -A' \dfrac{x_1}{x_1 + C_1}; & 0.1 \leq \theta < 1 \text{ (Mixed)} \\[2mm] -A'' \dfrac{x_1}{x_1 + C_1}; & 0 \leq \theta < 0.1 \text{ (Insoluble)} \end{cases} \qquad (\text{Eq. } 17)$$

The coefficients A, A′ and A″ are defined as follows, where the subscripts refer to the colliding modes (see Eq. 15 for details):

$$A = \rho_1 \frac{\pi}{6} [f_{1,4} + f_{1,7}],$$

$$A' = \rho_1 \frac{\pi}{6} [f_{1,3} + f_{1,5} + f_{1,6} + f_{1,8} + f'_{2,1}],$$

$$A'' = \rho_1 \frac{\pi}{6} [f_{1,3} + f_{1,6} + f_{1,9}].$$

The negative sign present across all cases of $\theta$ in Eq. 17 reflects that any intermodal coagulation event involving a particle in the ks mode will always result in its conversion to other aerosol modes. Applying Eq. 9 to Eq. 17 results in the following expression for the contribution of aviation to mode ks via coagulation:

$$\frac{\partial x_1^{\text{avi}}}{\partial t}\Big|_{\text{coag}} = F_1(\mathbf{x}) \frac{\mathbf{x}^{j^T} \nabla F_1(\mathbf{x})}{\mathbf{x}^T \nabla F_1(\mathbf{x})}.$$

Evaluating the sensitivity fraction leads to the final expression for $\frac{\partial x_1^{\text{avi}}}{\partial t}\Big|_{\text{coag}}$, where K = {A, A′, A″}:

$$\frac{\partial x_1^{\text{avi}}}{\partial t}\Big|_{\text{coag}} = F_1(\mathbf{x}) \frac{\left(x_1^{\text{avi}}, x_2^{\text{avi}}, x_3^{\text{avi}}, \ldots, x_9^{\text{avi}}\right)\left(\frac{-KC_1}{(x_1+C_1)^2}, 0, 0, \ldots, 0\right)^T}{(x_1, x_2, x_3, \ldots, x_9)\left(\frac{-KC_1}{(x_1+C_1)^2}, 0, 0, \ldots, 0\right)^T} = F_1(\mathbf{x}) \frac{x_1^{\text{avi}}}{x_1}. \qquad (\text{Eq. } 18)$$



Based on the result from Eq. 18, the tagging ratio $\frac{x_1^{avi}}{x_1}$ for the coagulation tendency of $SO_4$ in mode ks according to Grewe

(2013) is the ratio of soluble Aitken mode sulfate, $C_{SO_4,ks}\big|^{avi}$, to the total amount of this aerosol across all other sources, i.e.,

$\frac{x_1^{avi}}{x_1} = \frac{C_{SO_4,ks}\big|^{avi}}{C_{SO_4,ks}\big|^{avi}+C_{SO_4,ks}\big|^{rem}}$. Typically, the form of this ratio varies with $\theta$, but for mode ks, the same structure is obtained

(only K varies). This result is verified analytically as Eq. 10 is upheld. The derivation for the remaining eight aerosol modes is

included in Appendix C.

We highlight that this derivation considers the following simplifying assumptions:

1.    Coagulation kernels $\beta$ and terms $C_p$ are independent of the state variables $x_i$.
       2.    Coagulation between modes $p$ and $q$ leads to the same outcome as the coagulation between modes $q$ and $p$.

Given the complexity of the tagging ratios for the remaining modes resulting from the extra computational effort of storing

and passing all of the coagulation kernels across submodels and the added difficulty in implementing $\theta$-dependent piecewise

functions (e.g. Eq. C2 in Appendix C), we consider the method applied by IW17 in which secondary particulate sulfate ($SO_4$

that is indirectly produced from the oxidation of $SO_2$) production from source $j$ is tagged as a function of the emitted $SO_2$ (i.e.,

tagging ratio is $\frac{SO_2|^j}{\sum SO_2}$). In this study, both $SO_2$ and $H_2SO_4$ are precursors of particulate sulfate that are directly emitted and

tracked by AIRTRAC. Since $SO_2$ must first be oxidized to $H_2SO_4$ before being converted to $SO_4$, our tagging ratio involves

the latter:

$$\frac{\partial C_{SO_4,M}}{\partial t}\bigg|_{coag}^{avi} = \frac{C_{H_2SO_4}\big|^{avi}}{C_{H_2SO_4}\big|^{avi} + C_{H_2SO_4}\big|^{rem}} \times \frac{\partial C_{SO_4,M}}{\partial t}\bigg|_{coag}. \qquad (\text{Eq. }19)$$

Lastly, the growth and aging processes, represented by the "rename" tendency, are formulated analogously to the coagulation

(Eq. 18) and condensation (Eq. 14) tendencies, as these processes similarly depend on coagulation and condensation

mechanisms. The aviation-attributable growth and aging contribution is therefore written as:

$$\frac{\partial C_{SO_4,M}}{\partial t}\bigg|_{gr}^{avi} + \frac{\partial C_{SO_4,M}}{\partial t}\bigg|_{ag}^{avi} = \frac{\partial C_{SO_4,M}}{\partial t}\bigg|_{rename}^{avi} = \frac{C_{H_2SO_4}\big|^{avi}}{C_{H_2SO_4}\big|^{avi} + C_{H_2SO_4}\big|^{rem}} \times \frac{\partial C_{SO_4,M}}{\partial t}\bigg|_{rename}. \qquad (\text{Eq. }20)$$

### 3.5 Aviation-attributable $SO_4$ removal processes

The removal of aviation-produced $SO_4$ through the three processes described in Section 2.4 (dry deposition, sedimentation,

and scavenging) is scaled by the aviation-attributable $SO_4$ tagging ratio. This reflects the fact that removal processes are directly



proportional to the pollutant to be removed and is consistent with the method of IW17. In other words, the removal rate should be linearly proportional to the amount of the aerosol present in the atmosphere as the removal tendency should be approximately zero once the aviation-induced aerosols are nearly depleted. The dry deposition (DDEP), sedimentation (SEDI) and scavenging (SCAV) tendencies for aviation aerosols are therefore defined by Eq. 21:

$$\left.\frac{\partial C_{SO_4,M}}{\partial t}\right|_i^{avi} = \frac{\left.C_{SO_4}\right|^{avi}}{\left.C_{SO_4}\right|^{avi} + \left.C_{SO_4}\right|^{rem}} \times \left.\frac{\partial C_{SO_4,M}}{\partial t}\right|_i \;; i = \{DDEP, SEDI, SCAV\}. \tag{Eq. 21}$$

### 3.6 An overview of the tagging equations for aviation SO$_4$

By combining the tagging formulations for all aerosol microphysical and removal processes, we provide a consolidated overview of the non-linear tagging differential equations as a function of process tendencies to track the transport of aviation SO$_4$ across all nine modes. Equation 22 applies only to mode ks, while Eq. 23 to the remaining modes. The isotropic turbulent mixing parameterization from LGTMIX is represented by $R(C_{SO_4,M})$.

$$\left.\frac{\partial C_{SO_4,ks}}{\partial t}\right|^{avi} = \frac{\left.C_{H_2SO_4}\right|^{avi}}{\left.C_{H_2SO_4}\right|^{avi} + \left.C_{H_2SO_4}\right|^{rem}} \times \left( \underbrace{\left.\frac{\partial C_{SO_4,ks}}{\partial t}\right|_{nucl}}_{\text{Nucleation}} + \underbrace{\left.\frac{\partial C_{SO_4,ks}}{\partial t}\right|_{cond}}_{\text{Condensation}} + \underbrace{\left.\frac{\partial C_{SO_4,ks}}{\partial t}\right|_{coag}}_{\text{Coagulation}} + \cdots \right.$$

$$\left. \cdots + \underbrace{\left.\frac{\partial C_{SO_4,ks}}{\partial t}\right|_{rename}}_{\text{Growth \& Aging}} \right) - \frac{\left.C_{SO_4}\right|^{avi}}{\left.C_{SO_4}\right|^{avi} + \left.C_{SO_4}\right|^{rem}} \times \left( \underbrace{\left.\frac{\partial C_{SO_4,ks}}{\partial t}\right|_{DDEP}}_{\text{Dry Deposition}} + \underbrace{\left.\frac{\partial C_{SO_4,ks}}{\partial t}\right|_{SEDI}}_{\text{Sedimentation}} + \underbrace{\left.\frac{\partial C_{SO_4,ks}}{\partial t}\right|_{SCAV}}_{\text{Scavenging}} \right) \cdots \tag{Eq. 22}$$

$$\cdots + \underbrace{R(C_{SO_4,ks})}_{\text{Turbulence}}$$

Lastly, for the remaining modes $M = \{km, ki, as, am, ai, cs, cm, ci\}$:

$$\left.\frac{\partial C_{SO_4,M}}{\partial t}\right|^{avi} = \frac{\left.C_{H_2SO_4}\right|^{avi}}{\left.C_{H_2SO_4}\right|^{avi} + \left.C_{H_2SO_4}\right|^{rem}} \times \left( \underbrace{\left.\frac{\partial C_{SO_4,M}}{\partial t}\right|_{cond}}_{\text{Condensation}} + \underbrace{\left.\frac{\partial C_{SO_4,ks}}{\partial t}\right|_{coag}}_{\text{Coagulation}} + \underbrace{\left.\frac{\partial C_{SO_4,ks}}{\partial t}\right|_{rename}}_{\text{Growth \& Aging}} \right) + \cdots \tag{Eq. 23}$$



$$\dots - \frac{C_{SO_4}\big|^{avi}}{C_{SO_4}\big|^{avi} + C_{SO_4}\big|^{rem}} \times \left( \underbrace{\frac{\partial C_{SO_4,M}}{\partial t}\bigg|_{DDEP}}_{\text{Dry Deposition}} + \underbrace{\frac{\partial C_{SO_4,M}}{\partial t}\bigg|_{SEDI}}_{\text{Sedimentation}} + \underbrace{\frac{\partial C_{SO_4,M}}{\partial t}\bigg|_{SCAV}}_{\text{Scavenging}} \right) + \underbrace{R(C_{SO_4,M})}_{\text{Turbulence}}.$$

As it would be much more computationally demanding to develop a complete chemistry mechanism within every air parcel defined in a simulation, AIRTRAC leverages the background chemistry from the EMAC model by linearly scaling these non-linear background responses with the emission amount. Although this approach introduces simplifications like discarding 550 feedback effects between the emission and background, the computational requirements are decreased by at least one to two

orders of magnitude (Maruhashi et al., 2024). To linearize Eqs. 22 and 23, the tagging ratios $\frac{C_{H_2SO_4}\big|^{avi}}{C_{H_2SO_4}\big|^{avi} + C_{H_2SO_4}\big|^{rem}}$ and

$\frac{C_{SO_4}\big|^{avi}}{C_{SO_4}\big|^{avi} + C_{SO_4}\big|^{rem}}$ must be linearized. As it is comparable to the rational function $f(x) = \frac{x}{x+k}$, where $k$ is a real number, the

tagging ratios may be approximated with a first-order McLaurin polynomial: $\frac{x}{x+k} \approx \frac{x}{k} \Longrightarrow \frac{C_{H_2SO_4}\big|^{avi}}{C_{H_2SO_4}\big|^{avi} + C_{H_2SO_4}\big|^{rem}} \approx \frac{C_{H_2SO_4}\big|^{avi}}{C_{H_2SO_4}\big|^{rem}}$

and $\frac{C_{SO_4}\big|^{avi}}{C_{SO_4}\big|^{avi} + C_{SO_4}\big|^{rem}} \approx \frac{C_{SO_4}\big|^{avi}}{C_{SO_4}\big|^{rem}}$. The set of linearized tagging equations implemented in AIRTRAC v2.0 is therefore 555 represented by Eqs. 24 and 25:

$$\frac{\partial C_{SO_4,ks}}{\partial t}\bigg|^{avi} = \frac{C_{H_2SO_4}\big|^{avi}}{C_{H_2SO_4}\big|^{rem}} \times \left( \underbrace{\frac{\partial C_{SO_4,ks}}{\partial t}\bigg|_{nucl}}_{\text{Nucleation}} + \underbrace{\frac{\partial C_{SO_4,ks}}{\partial t}\bigg|_{cond}}_{\text{Condensation}} + \underbrace{\frac{\partial C_{SO_4,ks}}{\partial t}\bigg|_{coag}}_{\text{Coagulation}} + \underbrace{\frac{\partial C_{SO_4,ks}}{\partial t}\bigg|_{rename}}_{\text{Growth \& Aging}} \right) + \dots$$

(Eq. 24)

$$\dots - \frac{C_{SO_4}\big|^{avi}}{C_{SO_4}\big|^{rem}} \times \left( \underbrace{\frac{\partial C_{SO_4,ks}}{\partial t}\bigg|_{DDEP}}_{\text{Dry Deposition}} + \underbrace{\frac{\partial C_{SO_4,ks}}{\partial t}\bigg|_{SEDI}}_{\text{Sedimentation}} + \underbrace{\frac{\partial C_{SO_4,ks}}{\partial t}\bigg|_{SCAV}}_{\text{Scavenging}} \right) + \underbrace{R(C_{SO_4,ks})}_{\text{Turbulence}}.$$

For the remaining modes $M = \{km, ki, as, am, ai, cs, cm, ci\}$:

$$\frac{\partial C_{SO_4,M}}{\partial t}\bigg|^{avi} = \frac{C_{H_2SO_4}\big|^{avi}}{C_{H_2SO_4}\big|^{rem}} \times \left( \underbrace{\frac{\partial C_{SO_4,M}}{\partial t}\bigg|_{cond}}_{\text{Condensation}} + \underbrace{\frac{\partial C_{SO_4,ks}}{\partial t}\bigg|_{coag}}_{\text{Coagulation}} + \underbrace{\frac{\partial C_{SO_4,ks}}{\partial t}\bigg|_{rename}}_{\text{Growth \& Aging}} \right) + \dots$$

(Eq. 25)





$$... - \left.\frac{C_{SO_4}\big|^{avi}}{C_{SO_4}\big|^{rem}}\right. \times \left( \underbrace{\left.\frac{\partial C_{SO_4,M}}{\partial t}\right|_{DDEP}}_{\text{Dry Deposition}} + \underbrace{\left.\frac{\partial C_{SO_4,M}}{\partial t}\right|_{SEDI}}_{\text{Sedimentation}} + \underbrace{\left.\frac{\partial C_{SO_4,M}}{\partial t}\right|_{SCAV}}_{\text{Scavenging}} \right) + \underbrace{R\left(C_{SO_4,M}\right)}_{\text{Turbulence}}.$$

## 4 Application and evaluation of AIRTRAC v2.0

This section presents the application of AIRTRAC v2.0 in examining the transport patterns of aviation-emitted $SO_2$ and $H_2SO_4$ and their role in secondary $SO_4$ formation by comparing two emission points for the July – September 2015 emission scenario. The impact of seasonal shifts to the lifetimes of $SO_2$, $SO_4$ and to the mean productive efficiency of $SO_4$ is also considered. To 560   identify which of these emission points is most likely to lead to interactions with liquid clouds, ESA satellite cloud data (Stengel et al., 2019) is integrated into the analysis. Lastly the spatial distributions of $SO_4$ volume mixing ratios (VMRs) are compared to output from MADE3 using a perturbation approach and to results from Righi et al. (2023). While the absolute magnitudes obtained from the tagging and perturbation methods are not directly comparable—since they address fundamentally different research questions (Clappier et al., 2017)—their spatial distribution patterns are expected to be more comparable.

### 4.1 Analysis of directly emitted species: $SO_2$ and $H_2SO_4$

Figure 4 illustrates the differing transport pathways and impacts of $SO_2$ emissions originating from emission points 8 and 10, situated in the North and South Atlantic regions, respectively (see Fig. 2). Analogously, Fig. 5 presents the same analysis for $H_2SO_4$ for the same emission points. Both figures pertain to a 90 d period during the northern winter (July – September, 2015).

### 4.1.1 $SO_2$

In AIRTRAC v2.0, $SO_2$ is a tracer that undergoes a pure loss process, as was described by Eq. 11. Figures 4a and b depict the temporal evolution of $SO_2$ production efficiency and its e-folding time for two emission points, calculated by solving for $\tau$ using the exponential decay law: $C(t) = C_0 \cdot e^{-\frac{t}{\tau}}$, where $C_0$ is the initial concentration and $t$ is the time elapsed since emission. The mean e-folding time at point 8, approximately 27 d, is about 60% larger than the 17 d estimate for emission point 10. This disparity is likely due to the faster downward transport witnessed at point 10 (Fig. 4d), which accelerates $SO_2$ depletion from 575   larger background OH concentrations in the lower troposphere as a result of increased water vapor levels (Riedel and Lassey, 2008). The magnitudes of these $SO_2$ e-folding lifetimes ($\tau$) are in reasonable agreement with past studies that estimated e-folding lifetimes ranging from a few days (Beirle et al., 2014) to two weeks in the troposphere (von Glasow et al., 2009). We find that the median $SO_2$ lifetime between July – September is 14 d (Fig. 7a). $SO_2$ lifetimes will ultimately vary according to the drastically different depletion rates in the atmosphere (Oppenheimer et al., 1998; Beirle et al., 2014). In contrast, $SO_2$ 580   emitted at point 8 remains at higher altitudes for longer, where the stability and reduced $HO_x$ ($OH+HO_2$) species in the upper





atmosphere leads to slower depletion rates (Fig. 4c). Similar conclusions have been reported in previous studies for other chemical species, particularly in the context of $NO_x$ emissions affecting atmospheric $O_3$ concentrations, which are likewise oxidized by OH (Frömming et al., 2012; Rosanka et al., 2020; Frömming et al., 2021; Maruhashi et al., 2022).

The horizontal distribution associated with emission point 8 (Fig. 4e) shows that air parcels predominantly remain within the Northern Hemisphere, which agrees with previous findings noting that transhemispheric transport is limited (Grewe et al., 2002; Maruhashi et al., 2022). $SO_2$ emitted at point 10 mainly travels along the southern tropical region, with the largest scaled volume mixing ratios found along lower altitudes near the Equator. The tropical easterlies or trade winds typically acting between 0° and 30° S are the driving force behind the horizontal transport of $SO_2$ and carry emissions far to the West. The $SO_2$ production efficiency time series for all 28 emission points for January – March and July – September in 2015 are shown in 590 Figs. S1 and S2 in the Supplement (Maruhashi et al., 2025b) respectively.

**4.1.2 $H_2SO_4$**

$H_2SO_4$ volume mixing ratios along air parcel trajectories are calculated by AIRTRAC v2.0 using Eq. 12, whereby the conversion of $SO_2$ directly contributes to the formation of $H_2SO_4$. Consequently, the longer e-folding time for $SO_2$ at emission point 8 (Fig. 4a) also translates to a prolonged perturbation lifetime for $H_2SO_4$. Figure 5a indicates that $H_2SO_4$ persists beyond 595 the 80 d mark, whereas emissions from point 10 are fully depleted around 60 d after emission (Fig. 5b). As was observed by Rosanka et al. (2020) and Frömming et al. (2021) in the context of $NO_x$-$O_3$ chemistry, the air parcels that have an early initial descent to more chemically active regions in the lower troposphere tend to produce the largest maximum production efficiencies. This phenomenon is also present in Fig. 5b, where the largest $H_2SO_4$ production efficiency occurs for the air parcels from emission point 10 that exhibit an early descent.

The dependence on $SO_2$ means that $H_2SO_4$ production can continue to increase and even peak several days after the initial $SO_2$ emission, often occurring at locations far from the original source. Figures 5c and d exemplify this pattern as the maximum $H_2SO_4$ production efficiencies occur at much lower altitudes compared to the emission altitude. In terms of the horizontal distributions, Fig. 5e suggests that $SO_2$ emitted in the North Atlantic will lead to maximum $H_2SO_4$ production in parts of Africa and Asia. For an $SO_2$ emission in the South Atlantic (Fig. 5f), $H_2SO_4$ production is at a maximum in Central Africa and in 605 South America, resulting from the trade winds transporting the $SO_2$ to the West. The $H_2SO_4$ production efficiency time series for all 28 emission points for January – March and July – September in 2015 are shown in Figs. S3 and S4 in the Supplement (Maruhashi et al., 2025b) respectively.





Figure 4 – The spatio-temporal variation of aviation-emitted sulfur dioxide ($SO_2$) scaled by the emission mass for emission points 8 and 10 (see Fig. 2). Panels (a) – (b) display the temporal evolution of the $SO_2$ production efficiency throughout the 90 d simulation period (July – September, 2015). The multi-colored lines denote the production efficiencies across 50 air parcel trajectories initialized at the selected emission points. The thicker dark blue curve is the mean of these 50 trajectories. The values of $\tau$ represent the e-folding times in days for this mean curve. Panels (c) – (d) present the spatial variation of the production efficiency as a function of the pressure altitude and latitude. Panels (e) – (f) illustrate the spatial variation of the



production efficiency as a function of latitude and longitude. The green triangles indicate the approximate location of emission points 8 and 10, respectively.

Figure 5 – The spatio-temporal variation of aviation-emitted sulfuric acid (H₂SO₄) scaled by the emission mass for emission points 8 and 10 (see Fig. 2). Panels (a) – (b) display the temporal evolution of the H₂SO₄ production efficiency throughout the 90 d simulation period (July – September, 2015). The multi-colored lines denote the production efficiencies across 50 air parcel trajectories initialized at the selected emission points. The thicker dark blue curve is the mean of these 50 trajectories. The



values of $\tau$ represent the e-folding times in days for this mean curve. Panels (c) – (d) present the spatial variation of the production efficiency as a function of the pressure altitude and latitude. Panels (e) – (f) present the spatial variation of the production efficiency as a function of latitude and longitude. The green triangles show the approximate location of emission points 8 and 10, respectively.

## 4.2 Analysis of secondary SO$_4$

The amount of sulfate attributable to aviation emissions of SO$_2$ and H$_2$SO$_4$ is calculated according to Eqs. 24 and 25. Figure 6 displays the spatio-temporal evolution of total sulfate, which is defined as the sum of SO$_4$ across the nine aerosol modes mentioned in Section 2.3: $SO_4 = \sum_{i=1}^{9} SO_{4,i}$. On average, SO$_4$ production at emission point 10 (Fig. 6b) is nearly three times larger than at emission point 8 (Fig. 6a). This large discrepancy is owed to the significantly larger amount of H$_2$SO$_4$ that results from point 10, which is converted to SO$_4$ via processes like nucleation and condensation. Both Figs. 6a and b further exemplify this, as SO$_4$ production is initiated when H$_2$SO$_4$ production is maximal, which occurs approximately at the 30 d mark for point 8 (Fig. 5a) and at the 2 d mark for point 10 (Fig. 5b). As there is an approximate 20 d delay from the emission of SO$_2$ and H$_2$SO$_4$ until the point during which SO$_4$ production occurs, sulfate aerosols can form in regions far from the initial emission points, as was seen with H$_2$SO$_4$ in Fig. 5. The sulfate lifetimes approximate to 67 d and 64 d for points 8 and 10, respectively. Unlike precursor species such as SO$_2$, secondary sulfate exhibits significantly longer atmospheric lifetimes—ranging from several days to weeks in the troposphere (Textor et al., 2006; Boucher, 2015; Toohey et al., 2025), and extending to several months or even years in the stratosphere (Myhre et al., 2013; Sun et al., 2024; Toohey et al., 2025). The lifetimes estimated in this study are comparable to the upper end of the tropospheric range, likely due to the consideration of pulse emissions in the upper troposphere–lower stratosphere (UTLS), and fall well within the expected stratospheric range. A more detailed discussion of these calculations is provided in Section 4.3.

Regarding the spatial distribution of SO$_4$, flying at point 8 on that specific day is likely to lead to the strongest sulfate production in the Northern Hemisphere, between the latitudes of 30° and 60° N (Fig. 6c). According to the corresponding horizontal distribution panel (Fig. 6e), several regions beyond the local North Atlantic emission area are affected, including parts of Europe and Asia. For point 10, a wider latitudinal band is impacted, with the largest impacts noted between 0° and 60° S (Fig. 6d). The Pacific region close to the Equator is likely to experience the largest SO$_4$ production (Fig. 6f). The Brewer-Dobson Circulation (BCD) patterns also impact the vertical transport of sulfate. As has been noted by Sun et al. (2023), particles injected in the Tropics at around the upper troposphere and lower stratosphere closer to the Equator will generally experience upwelling and may be transported higher into the stratosphere via the deep branch of the BCD. This phenomenon is clearly observed in Figs. 6c and d, where air parcels rise above the emission point (green triangle) and reach the highest altitudes near 0º N. The highest point reached for an emission starting at point 8 is 103 hPa (~16.1 km) and for point 10 is 94 hPa (~16.7 km). Additionally, as emission point 10 is closer to the Equator, more sulfate aerosols are lofted into the stratosphere, which enhances sulfate VMRs at higher altitudes and may even prolong the overall lifetime of aerosols if they remain in the stratosphere.








Figure 6 – The spatio-temporal variation of aviation-emitted total sulfate (sum of nine modes of SO$_4$) scaled by the emission mass for emission points 8 and 10 (see Fig. 2). Panels (a) – (b) display the temporal evolution of the SO$_4$ production efficiency throughout the 90 d simulation (July – September, 2015). The multi-colored lines denote the production efficiencies across 50 air parcel trajectories initialized at the selected emission points. The thicker dark blue curve is shown at the top right corner.
The e-folding time ($\tau$) in days is shown in red and corresponding exponential lifetime fits are shown by the dashed red curves.





Panels (c) – (d) present the spatial variation of the production efficiency as a function of the pressure altitude and latitude. Panels (e) – (f) illustrates the spatial variation of the production efficiency as a function of latitude and longitude. The green triangles indicate the approximate location of emission points 8 and 10, respectively.

### 4.3 Seasonal effects

The production of sulfate aerosols from aviation exhibits a strong seasonal dependence as a result of variations in background chemistry and available solar radiation. Seasonal changes in the background water vapor and $HO_x$ levels significantly influence the oxidation pathways that eventually convert $SO_2$ into $SO_4$. As with most tropospheric compounds, the $SO_2$ lifetime depends on the OH radical, which drives its gas-phase oxidation. OH is prevalent in warm and humid locations as its formation relies on the photodissociation of $O_3$ and subsequent reaction with $H_2O$ (Riedel and Lassey, 2008). Figure 7a depicts the range of

$SO_2$ e-folding times for both seasons. The median e-folding time of 22 d in January (winter) is nearly 60% larger than the median of 14 d in July (summer), which is expected, given the increased solar radiation during summer that contributes to increased OH and faster $SO_2$ depletion rates. Consequently, this enhanced oxidation of $SO_2$ results in larger $SO_4$ production, as shown in Fig. 7b, where its median production efficiency is 144% larger in July compared to January. The summertime $SO_2$ median lifetime of 14 d is within the upper limit from past studies (von Glasow et al., 2009), indicating that our wintertime

estimate of 22 d is slightly overestimated. However, our range of $SO_2$ e-folding estimates agrees with results of the modeling study by Zhu et al. (2022) and with the observations from the 2022 Hunga Eruption from the Infrared Atmospheric Sounding Interferometer (IASI) satellite instrument that also estimated a similar upper limit lifetime of 21.4 d (Sellitto et al., 2024). This may be explained by our simplification of not accounting for the rapid aqueous-phase oxidation in the troposphere and scavenging processes, which act as significant $SO_2$ sinks. This explains the longer lifetimes of $SO_2$ in the stratosphere as wet

removal is less frequent, and $HO_x$ species are much less abundant (Brodowsky et al., 2021). Due to the skewed nature of the distributions shown in Figure 7, a non-parametric statistical approach is more appropriate than parametric alternatives that assume normality. Therefore, the Mann-Whitney U test (Mann and Whitney, 1947) was applied to assess the statistical significance of seasonal differences. Based on this test, both the $SO_2$ lifetime (p-value $= 3.57 \times 10^{-4}$) and the $SO_4$ mean production efficiencies (p-value $= 6.83 \times 10^{-3}$) are found to be statistically significantly larger at the 95% confidence level

during the winter and summer seasons, respectively.

The lifetime of sulfate (Fig. 7c) has a median close to 2 months, with a value of around 62 d in January and 65 d in July. Unlike $SO_2$, the median $SO_4$ lifetimes are not statistically significantly different across summer and winter according to the same test statistic and confidence interval mentioned earlier (p-value $= 0.55$). It is worth noting, however, that the maximum $SO_4$ lifetime during winter of 123.5 d is almost 30% larger than the maximum in summer (96.7 d). This causes the mean $SO_4$ lifetime to be

larger in January (69 d) than in July (67 d). The sulfate time series plots for all 28 emissions points for the periods January – March, 2015 and July – September, 2015 from which the e-folding times in Fig. 7 were calculated are shown in Figs. S5 and S6 in the Supplement (Maruhashi et al., 2025b).







Figure 7 – Seasonal comparison of (a) sulfur dioxide (SO$_2$) e-folding times (e.g. same $\tau$ values in Fig. 4), (b) the three-month mean sulfate (SO$_4$) production efficiencies and (c) the sulfate e-folding times. Outliers in (b) are not shown for clarity. The horizontal axes describe the month of emission where January denotes the period January – March, 2015 and July denotes July – September, 2015.

Our range of calculated SO$_4$ lifetimes is consistent with output from another Lagrangian passive tracer pulse experiment from Toohey et al. (2025) that employs the FLEXPART model. In their analysis, stratospheric sulfate aerosol transport and lifetimes were analyzed in the context of volcanic SO$_2$ eruptions in the Northern Hemisphere. The altitudinal range of tracer injections in December and June that they considered varies from 13 – 25 km, where SO$_4$ aerosol lifetimes increased sharply with altitude. Since we emit SO$_2$ pulses at the UTLS that lead to sulfate maxima reaching around 100 hPa (~16 km) according to Figs. 6c and d, our results will also be comparable to theirs. According to their Fig. 4, sulfate aerosols at an altitude of 13 km between 40° and 60° N exhibit lifetimes ranging from 1.9 to 3.9 months during June (summer) and from 3.4 to 4 months during December (winter). Our Fig. 7c displays a median value of 62 d or ~2.1 months for January and ~2.2 months for July. Our



estimates are expectedly lower than their estimated range given our lower emission altitude. Furthermore, seeing as there is a portion of sulfate that reaches the stratosphere at around 16 km in altitude (e.g. Fig. 6), upper bound values in Fig. 7c such as 96.7 d (~3.2 months) in July and 123.5 d (~4.1 months) in January are also justified bearing in mind that, Sun et al. (2023), for instance, for an injection height of 16 km, found particle lifetimes throughout the year that range from 2.4 to 9.6 months, which are similar to both of our seasonal estimates from Fig. 7c. The lower stratospheric aerosol lifetime estimate by Sun et al. (2024)

estimated a smaller stratospheric aerosol lifetime of 4.8 months for a 65 hPa pressure altitude (~18.5 km). By contrast, particle lifetimes according to Toohey et al. (2025) for an injection height of 15 km were larger, ranging from 6.1 – 7.5 months. Differences between estimates from AIRTRAC and other models like LAGRANTO are, however, naturally expected as the latter does not consider aerosol microphysical processes like particle growth along trajectories.

## 4.4 Application to aerosol-cloud interactions

To demonstrate the capability of AIRTRAC v2.0 in predicting when aviation-attributable sulfate aerosols will likely interact with low-level liquid clouds, we incorporate satellite data from ESA's Climate Change Initiative (CCI) to identify the approximate locations of these clouds. More specifically, version 3 of the Advanced Very High Resolution Radiometer ante meridiem dataset (AVHRR-AMv3; Stengel et al., 2019) at monthly means is used, which covers the period from 1982 to 2016. This comprehensive dataset comprises 174 variables with a latitude-longitude grid resolution of 0.5°×0.5°. To pinpoint the

location of liquid clouds during the simulated periods in summer and winter, the liquid cloud fraction (LCF) and cloud top pressure (CTP) variables were analyzed.

Based on Fig. D1 from Appendix D, liquid clouds are predominantly found at CTPs below 800 hPa and are primarily concentrated in the Northern Atlantic and Southern Tropics, with a notably larger mean LCF observed during the summer months (July – September, 2015). During this period, liquid clouds are predominantly found between 0º - 30º S across all

longitudes. These horizontal liquid cloud distributions align with those reported by Stengel et al. (2020) for June 2014. Similarly, Lauer et al. (2007), using annual mean satellite data from the International Satellite Cloud Climatology Project (ISCCP) for the period 1983 – 2004, confirm a significant presence of low-level liquid clouds in the North Atlantic, North Pacific and South Atlantic regions.

Based on observations from this cloud satellite dataset, the region most likely to form low-level liquid clouds between July –

September, 2015 is marked with a blue rectangle in Fig. 8. When emissions occur near the Northern Mid-latitudes during summer (Fig. 8a), $SO_4$ may have an extended residence time near the cruise altitude, resulting in a slower downward transport to lower altitudes. The green median trajectory shows that air parcels following this path remain close to the emission pressure altitude of approximately 240 hPa for around 8 d before descending towards the surface, eventually reaching latitudes between 30º - 60º N. Although it is unlikely that $SO_4$ will reach the primary region with liquid clouds indicated in blue, it could still



interact with some clouds present in the North Atlantic (see Fig. D1). In contrast, emissions in the Southern Tropics (Fig. 8b) present an increased likelihood for interactions between aviation sulfate and liquid clouds in the Southern region.

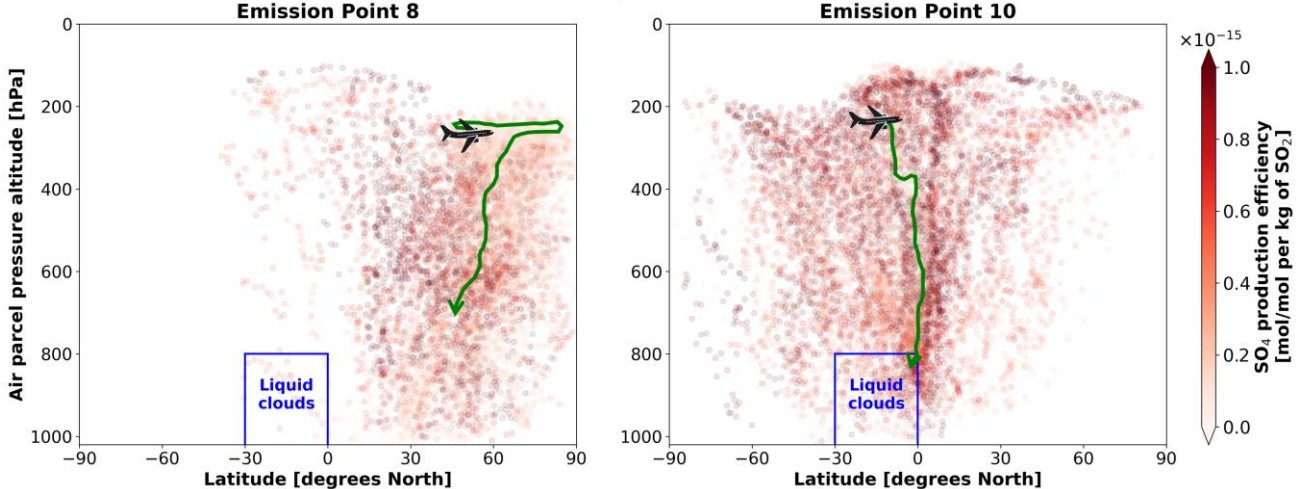

Figure 8 – Transport patterns for aviation-induced sulfate aerosols when SO$_2$ and H$_2$SO$_4$ are jointly emitted during July 2015 at (a) emission point 8 and at (b) emission point 9. The green curves represent the approximate median trajectories in each case. The aircraft icon indicates the approximate emission point. The blue region denotes the most probable location of lower-
level liquid clouds between July – September of 2015 according to the AVHRR-AMv3 dataset.

## 5 Evaluation of AIRTRAC v2.0

The conservation of sulfur species (closed mass budget) across the microphysical aerosol tendencies from MADE3 (growth, aging, coagulation, condensation and nucleation) that AIRTRAC v2.0 scales has been validated by Sharma et al. (2025). In general, the direct evaluation of tagging models using observational data is difficult, since observed concentrations of chemical
species are total quantities that emanate from various sources, both biogenic and anthropogenic, and disentangling these categories is challenging. Direct observational data for benchmarking are therefore unavailable. From a modeling perspective, evaluation is also challenging as many Lagrangian models are geared towards the stratospheric analysis of sulfate, as has been referred in earlier sections. However, the underlying components and submodels of AIRTRAC, like MADE3, have been extensively evaluated against a variety of experimental data (Kaiser et al., 2019). The lifetimes calculated by AIRTRAC v2.0
of precursor species like SO$_2$ and secondary SO$_4$ have also been shown to agree reasonably well in Sections 4.1 and 4.3 respectively with the output from other Lagrangian models applied to the study of stratospheric SO$_4$ from volcanic pulse emissions of SO$_2$ (Sun et al., 2023; Sun et al., 2024; Toohey et al., 2025). To evaluate the spatial distribution patterns of SO$_4$, a perturbation approach using a modeling setup similar to Righi et al. (2023) was applied to generate comparable simulation output. This simulation setup provides the most suitable basis for comparison due to its use of the same EMAC submodels and
emissions inventories (i.e. the same aviation SO$_2$ mass fluxes from CMIP6 for the year 2015), ensuring consistent background conditions. Additionally, for increased comparability, the CMIP6 aviation SO$_2$ emissions inventory applied in the perturbation





method has been modified to only consider mass fluxes at the same pressure level of 240 hPa as is shown in Fig. 2. It is important to reiterate that while the spatial patterns of $SO_4$ can be meaningfully compared, the magnitudes from the tagging and perturbation approaches are not expected to match, as each method addresses fundamentally different research questions
(Clappier et al., 2017).

Figure 9 presents a comparison of the VMRs for $SO_4$ according to AIRTRAC v2.0 (Figs. 9a and c) and the perturbation approach (Figs. 9b and d) for both simulation periods. In AIRTRAC simulations, each of the 28 points (Fig. 2) is treated as an independent emission scenario, with no chemical interaction between any of the emissions considered (Maruhashi et al., 2024). Consequently, the values in Figs. 9a and c represent the total sulfate VMR across all 28 points and nine aerosol modes.
However, a direct comparison of magnitudes is still challenging, as the perturbation approach inherently accounts for background feedbacks (i.e. local saturation effects and depletion of oxidizing species like OH) and non-linear interactions across emissions. Furthermore, the nature of emissions is also different: 15-minute pulses are applied for AIRTRAC, and sustained emissions across three months are assumed for the perturbation method. Although a direct comparison of magnitudes is difficult, both methods yield estimates of the same order of magnitude, i.e., $10^{-12}$ mol·mol$^{-1}$.

Both methods reveal fairly similar transport patterns, highlighting a production lag of sulfate relative to $SO_2$ emissions. This is evident in the larger VMRs that appear at altitudes below the original emission level of 240 hPa. Both approaches also agree that the tropical latitudinal band between 0° and 20° N near the surface is most strongly affected, in line with the results of Righi et al. (2023) for total aviation $SO_4$ in 2015 (see their Fig. S15b). While both methods capture enhanced VMRs near the emission altitude (~200 hPa), the perturbation approach predicts stronger sulfate concentrations in the upper northern latitudes
(above 40° N; Figs. 9b,d) during both seasons, which is again consistent with Righi et al. (2023). AIRTRAC, by contrast, produces larger VMRs at this altitude closer to the northern tropics and mid-latitudes (Figs. 9a,c). This discrepancy may arise because the perturbation approach assumes a more widespread distribution of $SO_2$ given the sustained nature of its emissions, whereas AIRTRAC transports 15-min pulse emissions that are largest in magnitude over the mid-latitudes between 30° N and 40° N (e.g. emission points 4, 7, 15, and 18 in Fig. 2).

During winter, both approaches predict an enhancement of sulfate near the surface in the tropics (Fig. 9a and b). Close to a pressure altitude of 400 hPa, the perturbation approach estimates considerably smaller VMRs at the tropics. This could arise from feedbacks that mask effects (similar to chemical applications, see Grewe et al., 2012) or from the consideration that 10 out of 28 emission points are located below 30º N, even if individually they represent smaller emissions compared to, for instance, points 7 and 15 in Fig. 2. During summer, discrepancies between the approaches are larger, both near the surface and
close to altitudes between 400 and 500 hPa (Figs. 9c and d). However, the region with the largest VMRs occurs below 600 hPa for both models and in the latitudinal band between 0º – 40º N. At the lowermost layers, AIRTRAC estimates negligible $SO_4$ VMRs, which could be due to the inherent limitations from the Lagrangian advection scheme itself, performed by the ATTILA submodel (Reithmeier and Sausen, 2002; Brinkop and Jöckel, 2019). As the grid-point equivalent VMRs are





calculated based on the average VMRs across all air parcels in a grid box, and the distribution of air parcels is not strictly

always representative of the pressure and air density distribution in the grid space, the mapping between Lagrangian space and Eulerian frames will lead to differences in the calculation of VMRs of the air parcels (Grewe et al., 2014b). Additionally, AIRTRAC estimates are also conditioned by the number of air parcels per grid box. According to Reithmeier and Sausen (2002), the average number of air parcels near the surface for a T30 resolution is between 0.25 and 0.5 (see their Fig. 1). This could also explain the much smaller VMRs near the surface in Figs. 9a and c.

Another source for the discrepancies between the spatial patterns in Fig. 9 could be the linear parameterization for the mixing between air parcels by the LGTMIX submodel. The mixing intensity is controlled by two constants ($d_{trop} = 10^{-3}$; $d_{strat} = 10^{-6}$), one for each layer of the atmosphere. These were initially proposed by Collins et al. (1997) for gases and $d_{strat}$ was later changed to $5 \times 10^{-4}$ by Reithmeier and Sausen (2002). Although short-lived species have been shown to be less sensitive to these parameters (Reithmeier and Sausen, 2002), the suitability of these constants to model aerosol mixing within air parcels

has not been assessed. Further on the topic of linearization, AIRTRAC's linearly independent $SO_4$ contribution estimates for each emission point are added to yield the total $SO_4$ (includes all nine modes) field in Fig. 9. In contrast to the perturbation approach, this total neglects non-linear interaction effects between pulse emissions. For aviation $NO_x$–$O_3$ interactions, such non-linearities have the capability to alter production efficiencies by up to 30 % (Maruhashi et al., 2024). Ultimately, however, the sulfate spatial distributions between AIRTRAC and the perturbation approach largely agree that the lower altitudes between

the tropical to mid-latitudinal bands are the most affected. Despite methodological (tagging vs. perturbation) and emission type (pulse vs. sustained) discrepancies, both approaches still yield estimates within the same order of magnitude.





Figure 9 – Comparison of AIRTRAC's tagging estimates with the MADE3-based perturbation approach using a modeling setup similar to Righi et al. (2023) for total SO$_4$ (sum of nine aerosol modes) during the period July – September, 2015. Estimates from AIRTRAC v2.0 (panels (a) and (c)) represent the sum across all 28 emission points of the total SO$_4$ at each of these points.

## 6 Discussion and conclusion

### 6.1 Model advances, limitations and future research

AIRTRAC v2.0 is the first Lagrangian sulfate tagging scheme within EMAC, providing the unique capability to track the long-range atmospheric evolution of aviation-emitted sulfur compounds. Starting from the emission of SO$_2$ and H$_2$SO$_4$ at subsonic cruise altitudes, the scheme follows their chemical transformation into SO$_4$ followed by the subsequent descent of these



aerosols into the lower troposphere, where interactions with liquid clouds may occur. By considering aerosol microphysical processes like nucleation, condensation, coagulation, aging and particle growth, AIRTRAC v2.0 can calculate the VMRs across nine sulfate aerosol modes. It is therefore a useful computational tool that may help predict which flight regions are most likely to lead to aerosol-cloud interactions. The quantification of these interactions and how they impact the Earth's
radiative budget (i.e. in terms of radiative forcing) is a key next step.

In terms of limitations, the linearization of the production and loss tendencies from MADE3 in AIRTRAC discards climate feedback effects that could be introduced by the $SO_2$ and $H_2SO_4$ emissions. This means that locally, for instance, the atmosphere's oxidative capacity will not vary with the magnitude of the pulse emissions, meaning that background OH levels will remain unaffected by the amount of $SO_2$ introduced by AIRTRAC. The implications of this linearization have been
analyzed in greater detail in the context of aviation $NO_x$-$O_3$ interactions by Maruhashi et al. (2024). As has been introduced in Section 5, the suitability of air parcel mixing parameters that were originally formulated mostly for gas-phase species still needs to be better understood. As smaller aerosols in the Aitken mode are likely to behave more closely to gaseous species when mixing and larger particles in the coarse mode will not mix as rapidly given their much larger momenta, the introduction of size-dependent mixing parameters should be considered.

Another limitation of AIRTRAC v2.0 is that it cannot track sulfate particles that have been scavenged and later re-evaporated back into the atmosphere, which may lead to underestimations of aerosol mixing ratios. Another factor potentially contributing again to the underestimation of sulfate is the exclusion of its aqueous-phase production pathway via $SO_2$ oxidation by $H_2O_2$, due to the computational limitation discussed by Tost et al. (2006). This not only has implications for the amount of sulfate produced, but will also tend to overestimate $SO_2$ estimated lifetimes (Brodowsky et al., 2021). Longer $SO_2$ lifetimes may then
also lead to longer sulfate lifetimes. Excluding plume-scale processes can lead to an overestimation of around 15% of the aviation-induced sulfate particle number concentration, but has a negligible impact on sulfate mass when compared to an instant dispersion approach, which is adopted in AIRTRAC v2.0 (Sharma et al., 2025).

A few final considerations to further extend the capabilities of AIRTRAC v2.0 would be to firstly include the indirect impact of aviation $NO_x$ emissions on sulfate via the production of OH by connecting its gas-phase capabilities (version 1.0) to its new
aerosol capabilities. AIRTRAC could then provide a more comprehensive assessment of climate effects from aviation $NO_x$ effects, which will remain relevant even for drop-in sustainable aviation fuels and emerging hydrogen-powered aircraft (Tiwari et al., 2024). This could be contemplated by updating the tagging ratio of Eq. 11 based on the biomolecular reaction formulation of the tagging ratio, as was described earlier in Section 3.2:

$$C_{SO_2}\big|^{avi} = \underbrace{C_{SO_2}(t=0)}_{\text{Emission}} - \underbrace{\frac{1}{2} \times \left( \frac{C_{SO_2}\big|^{avi}}{C_{SO_2}\big|^{avi} + C_{SO_2}\big|^{rem}} + \frac{C_{OH}\big|^{avi}}{C_{OH}\big|^{avi} + C_{OH}\big|^{rem}} \right)}_{\text{Revised tagging ratio}} \times \underbrace{\frac{M_{SO_2}}{M_{H_2SO_4}}}_{\text{Molar masses}} \times \underbrace{P_{H_2SO_4}}_{\text{Production rate}} \times \Delta t + \underbrace{R\left(C_{SO_2}\right)}_{\text{Turbulence}}. \quad (Eq.\,26)$$



Secondly, to model the influence of aviation-induced sulfate on the microphysical properties of liquid clouds (e.g. by estimating the aerosol activation into cloud condensation nuclei), it would be necessary to extend the tagging from AIRTRAC v2.0 to also include the aerosol particle number concentration, as this is a key quantity driving aerosol-cloud interactions.

Lastly, AIRTRAC has been previously used as the computational foundation for the climate change functions (CCFs), which associate a change in temperature resulting from a local emission (Grewe et al., 2014a). Thus far, only CCFs for the net $NO_x$

effect, $H_2O$ and contrails have been formulated for a limited set of emission points in the Northern Trans-Atlantic (Frömming et al., 2021). By running AIRTRAC v2.0, we have generated a dataset to analyze the transport of aviation-induced $SO_2$, $H_2SO_4$ and nine aerosol modes containing $SO_4$. This dataset may later serve as the basis for the development of the first aerosol-cloud interaction CCFs, thereby furthering the possibility of applying them to climate-optimal routing (Sausen et al., 1994; Grewe et al., 2014a).

**6.2 Conclusion**

This paper presents the new functionalities and technical implementation of the AIRTRAC v2.0 submodel within the EMAC modeling framework. It applies a Lagrangian tagging approach to estimate the contributions of aviation-induced $SO_2$ and $H_2SO_4$ emissions to nine aerosol modes of atmospheric sulfate. To showcase AIRTRAC's capabilities in characterizing the transport patterns of aviation sulfur compounds, two three-month simulation scenarios (winter and summer) were performed

based on the CMIP6 emissions inventory. Furthermore, ESA satellite data were incorporated to demonstrate how AIRTRAC can predict where aviation-induced sulfate will reach lower-level liquid clouds.

The transport patterns of $SO_2$, $H_2SO_4$ and $SO_4$ are primarily driven by the atmospheric circulation, where trade winds and downdrafts from, e.g., subsidence events play a particularly important role. It was found, for instance, that $SO_2$ that is quickly transported to lower altitudes has a shorter atmospheric lifetime due to the locally larger background concentrations of OH,

which are responsible for oxidizing $SO_2$ as well as many other chemical species. For the same reason, $SO_2$ that is transported to lower altitudes early may lead to the largest sulfate production efficiency given the increased oxidative potential at lower tropospheric altitudes. This is consistent with past studies that have focused on $NO_x$ (Rosanka et al., 2020). Additionally, the BCD is also influential in determining the amount of sulfate that is lofted into the stratosphere, where its lifetime is longer. For example, species emitted from points closer to the Equator may be transported higher through the upwelling of the BCD

in the tropical region, which according to our results can reach pressure altitudes above 100 hPa. This phenomenon was also observed in another Lagrangian modeling study (Sun et al., 2023).

Seasonal variations were also found to impact the lifetime of $SO_2$ and the production efficiency of $SO_4$. The median lifetime of $SO_2$ during winter (January – March, 2015) was approximately 22 d compared to 14 d during the summer (July – September, 2015). The production efficiencies of $SO_4$ also vary seasonally given the dependence of OH concentrations on solar radiation:

in summer, increased sunlight enhances OH production, thereby boosting $SO_4$ formation. Overall, there was a 40% reduction




of $SO_2$ lifetimes in summer relative to winter and the $SO_4$ production efficiency in summer was 144% larger, these differences were found to be statistically significant at a 95% confidence level. Although median sulfate lifetimes were found to be larger in summer by almost 4%, the maximum $SO_4$ lifetime during winter was approximately 30% larger than in summer. The differences in $SO_4$ lifetimes were not found to be statistically significant.

We have also provided a case study in which AIRTRAC v2.0 would be useful to predict if $SO_2$ emissions from cruise would be transported far enough to reach the lower levels of the atmosphere that are more prone to liquid cloud formation, as indicated by ESA satellite cloud data. A clear difference was found between two emission scenarios: emitting in the usual North Atlantic Flight Corridor is unlikely to lead to significant aerosol interactions with the predominant liquid cloud cover in the South Atlantic, whereas flying at typical subsonic cruise levels (~240 hPa) in the Tropics would lead to a greater likelihood of aerosol-
cloud interactions.

AIRTRAC v2.0 relies largely on the aerosol tendencies from the MADE3 submodel, which have been evaluated extensively against observations from satellites, in-situ aircraft measurement and ground-based station data. We have validated AIRTRAC v2.0 output itself, however, with a combination of other Lagrangian modeling studies and with our results from a perturbation-based modeling approach. For example, lifetime estimates of $SO_2$ and $SO_4$ have been compared with both modeling and
observational studies of sulfate formation from volcanic eruptions, which can likewise be represented as a pulse emission of $SO_2$ at an above-ground injection altitude, followed by the production of $SO_4$ and its transport into the stratosphere. Given that some of our sulfate emissions also reach the stratosphere at similar injection heights tested by these studies, we have used them as reference. Our median $SO_2$ lifetimes agree mostly well with these past estimates ranging between a couple of days to three weeks. Our median $SO_4$ lifetimes of 2.1 and 2.2 months in January and July, respectively, likewise agree with ranges provided
by available studies. To assess the spatial distribution patterns of $SO_4$ predicted by AIRTRAC v2.0, we performed additional perturbation simulations with a comparable setup. Results showed that both approaches are consistent in identifying a similar latitudinal band in which sulfate VMR maxima are found in the lower troposphere. The near-surface estimates differ the most, where such discrepancies are likely owed to the limitations of a Lagrangian approach in having fewer air parcels near the surface (~0.25 – 0.5 per grid box) compared to higher parts of the atmosphere, where around 12 – 24 times as many air parcels
can exist per grid box.

Overall, AIRTRAC v2.0 is a promising tool that estimates lifetimes of sulfur-based species that drive aerosol-cloud interactions. Additionally, it can map the spatial distribution patterns to predict the emission points that are more likely to lead to impacts on lower-level liquid clouds. It is also highly efficient, capable of calculating up to around 28 emission scenarios in a single simulation, depending on the characteristics of the high-performance cluster.





## Appendix A – list of EMAC submodels applied

Table A1 – List of MESSy submodels that were used for the simulations in this study. A brief description of each one is provided along with a reference that can be consulted for further information.

| Submodel | Function | Source |
|---|---|---|
| AEROPT | Calculates aerosol optical properties. | Dietmüller et al. (2016) |
| AIRSEA | Computes exchange of chemical species between atmosphere and ocean. | Pozzer et al. (2006) |
| AIRTRAC | Calculates the contribution of a $NO_x$ or $H_2O$ emission to the atmospheric composition along air parcel trajectories. With version 2.0, the influence of $SO_2$ and $H_2SO_4$ on $SO_4$ may be studied. | Supplement of Grewe et al. (2014a), This study |
| ATTILA | Transport scheme for Lagrangian tracers. | Brinkop and Jöckel (2019) |
| CLOUD | Calculates cloud microphysics. | Kuebbeler et al. (2014) |
| CLOUDOPT | Calculates cloud optical properties. | Dietmüller et al. (2016) |
| CONVECT | Calculates convection based on different parameterizations. | Tost et al. (2006) |
| CVTRANS | Calculates transport of tracers due to convection. | Tost (2019) |
| DDEP | Calculates the dry deposition of gases and aerosols. | Kerkweg et al. (2006a) |
| E5VDIFF | Vertical diffusion for the ECHAM5 GCM. | Supplement of Emmerichs et al. (2021) |
| JVAL | Calculation of photolysis rates. | Sander et al. (2014) |
| LGTMIX | Parameterization for Lagrangian air parcel mixing. | Brinkop and Jöckel (2019) |
| LNOX | Parameterization for lightning $NO_x$ emissions. | Tost et al. (2007) |
| MADE3 | Calculation of aerosol microphysical processes. | Kaiser et al. (2019), Beer et al. (2020) |
| MECCA | Calculates tropospheric and stratospheric chemistry. | Sander et al. (2019) |
| OFFEMIS | Converts offline prescribed gridded emission fluxes from inventories to grid-point and Lagrangian tracer tendencies. | Kerkweg et al. (2006b) |
| ONEMIS | Calculates parameterized emission fluxes online during the simulation, including mineral dust and sea salt for aerosol emissions, and dimethyl sulfide (DMS) and nitric oxide (NO) for gaseous emissions. | Kerkweg et al. (2006b) |
| ORBIT | Calculates parameters related to Earth's solar orbit. | Dietmüller et al. (2016) |
| OROGW | Parameterization for orographic wave drag. | Roeckner et al. (2003) |
| SCAV | Wet scavenging process for gases, aerosols and liquid-phase chemistry. | Tost et al. (2006) |
| SEDI | Calculates sedimentation of aerosols. | Kerkweg et al. (2006) |
| SURFACE | Calculates surface temperatures over land and ocean. | MESSy Submodels (2024) |
| TNUDGE | Newtonian relaxation of user-defined tracers towards prescribed fields. | Kerkweg et al. (2006) |
| TREXP | Defines emission point sources for tracers. | Jöckel et al. (2010) |
| TROPOP | Meteorological diagnostics, such as tropopause height and planetary boundary layer height. | MESSy Submodels (2024) |






**Appendix B – determination of SO$_2$ and H$_2$SO$_4$ emission locations**

We provide, in Table B1, the coordinates of the 28 emission points in Fig. 2. These have been calculated by first identifying the pressure altitude for which the zonally averaged aviation SO$_2$ mass flux is the largest (Fig. B1). The distribution of latitude-longitude coordinates (Fig. B2) is obtained by selecting the maximum SO$_2$ fluxes from the 28 grid cells with the largest SO$_2$

flux contributions (Eq. B1). The EP_selector tool (see Maruhashi et al. (2025a) for the full code) has been coded in Python to systematically output the coordinates of these locations by computing flux contributions across all user-defined grid cells. For those with the largest SO$_2$ mass flux contributions, the coordinates of the maxima are recorded. As the EP_selector tool possesses several inputs like the latitudinal and longitudinal spacing for each grid cell, as well as the number of grid cells in which to search for the n largest values, it is necessary to select an appropriate combination for these parameters. Our chosen

combination is based on maximizing the total grid cells while ensuring that the total flux contribution is above 90%.

Table B1 – SO$_2$ and H$_2$SO$_4$ emission coordinates and amounts in kg. The contribution column is calculated according to Eq. B1 and represents the area-weighted mean mass flux of a grid cell. The altitude for all locations is 238.2 hPa.

| EP | Latitude [deg. North] | Longitude [deg. East] | Contribution | Emission [kg(SO$_2$)] | Emission [kg(H$_2$SO$_4$)] |
|----|----|----|----|----|----|
| 1 | 22.25 | -155.25 | 0.9% | 9.48E+03 | 2.96E+02 |
| 2 | 44.25 | -142.25 | 1.4% | 1.26E+04 | 3.93E+02 |
| 3 | 61.25 | -155.25 | 2.6% | 1.50E+04 | 4.69E+02 |
| 4 | 36.25 | -112.25 | 6.8% | 5.92E+04 | 1.85E+03 |
| 5 | 50.25 | -100.25 | 2.5% | 1.41E+04 | 4.40E+02 |
| 6 | 29.25 | -81.25 | 3.1% | 3.33E+04 | 1.04E+03 |
| 7 | 42.25 | -71.25 | 11.0% | 9.58E+04 | 2.99E+03 |
| 8 | 51.25 | -61.25 | 2.0% | 1.17E+04 | 3.65E+02 |
| 9 | -20.25 | -44.25 | 0.9% | 9.37E+03 | 2.93E+02 |
| 10 | -9.25 | -37.25 | 1.0% | 1.10E+04 | 3.43E+02 |
| 11 | 20.25 | -20.25 | 1.1% | 1.15E+04 | 3.59E+02 |
| 12 | 48.25 | -58.25 | 3.0% | 2.59E+04 | 8.09E+02 |
| 13 | 53.25 | -36.25 | 4.9% | 2.80E+04 | 8.75E+02 |
| 14 | 29.25 | -14.25 | 1.5% | 1.57E+04 | 4.90E+02 |
| 15 | 48.25 | 5.75 | 9.7% | 8.48E+04 | 2.65E+03 |
| 16 | 52.25 | -3.25 | 5.5% | 3.12E+04 | 9.76E+02 |
| 17 | 27.25 | 53.75 | 1.4% | 1.53E+04 | 4.79E+02 |
| 18 | 44.25 | 20.75 | 6.4% | 5.58E+04 | 1.74E+03 |
| 19 | 53.25 | 20.75 | 3.2% | 1.82E+04 | 5.67E+02 |
| 20 | 16.25 | 98.75 | 2.6% | 2.78E+04 | 8.67E+02 |
| 21 | 43.25 | 66.75 | 2.4% | 2.06E+04 | 6.43E+02 |
| 22 | 69.25 | 66.75 | 1.9% | 1.09E+04 | 3.42E+02 |
| 23 | 3.25 | 105.75 | 1.7% | 1.89E+04 | 5.91E+02 |
| 24 | 27.25 | 123.75 | 3.5% | 3.79E+04 | 1.18E+03 |
| 25 | 42.25 | 135.75 | 4.5% | 3.94E+04 | 1.23E+03 |
| 26 | 56.25 | 121.75 | 1.2% | 7.11E+03 | 2.22E+02 |
| 27 | 38.25 | 142.75 | 3.4% | 2.96E+04 | 9.24E+02 |
| 28 | 51.25 | 160.75 | 2.3% | 1.33E+04 | 4.14E+02 |



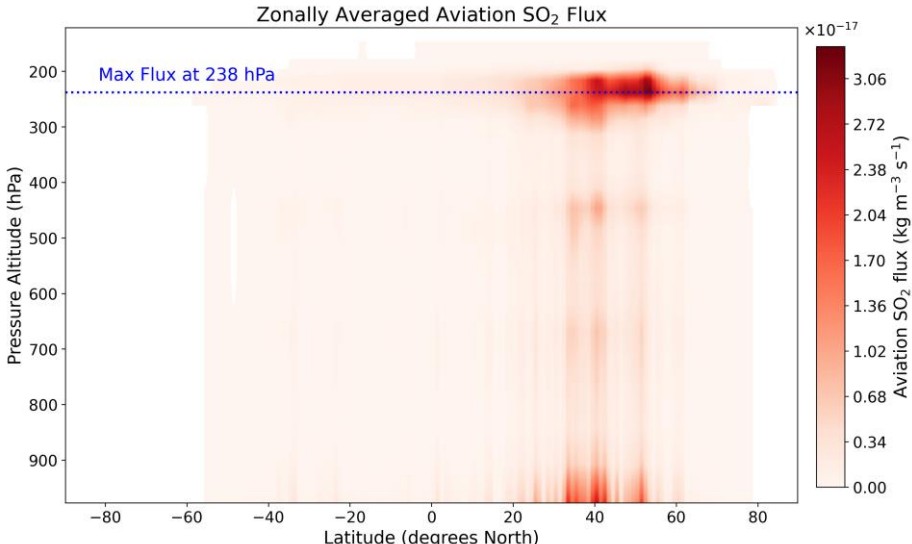

Figure B1 – Altitudinal variation of the zonally averaged aviation $SO_2$ mass flux [kg m$^{-3}$ s$^{-1}$] from the CMIP6 aviation emissions inventory. The blue dotted line represents the pressure altitude of 238.2 hPa (~10.6 km) at which the maximum flux occurs.

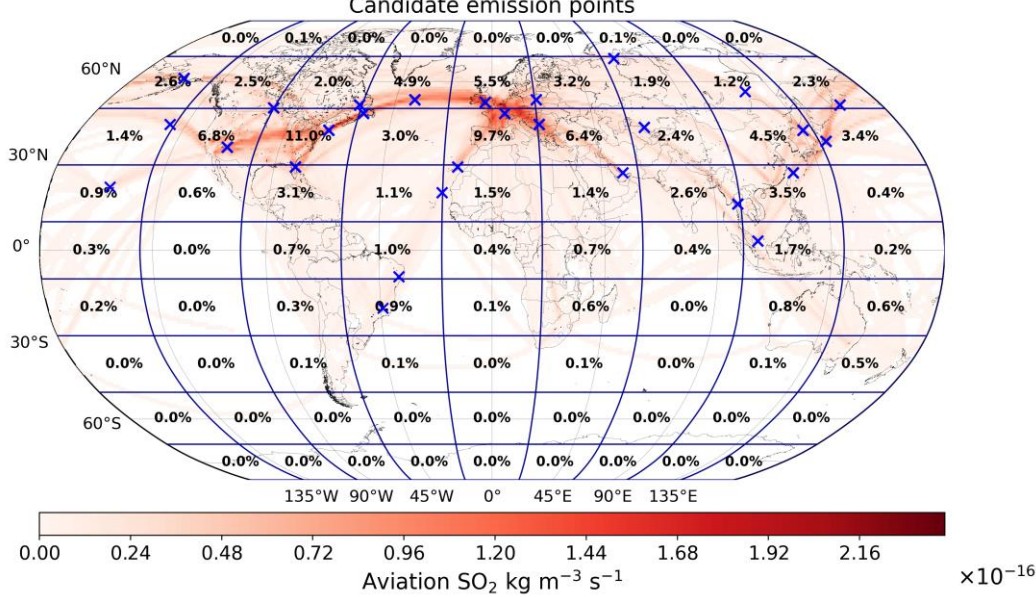

Figure B2 – Horizontal distribution of the 28 emission points (blue 'x' marks) determined by the EP_selector tool. The blue grid lines indicate the coarser user-defined mesh produced from spacings of 20º and 40º in the latitudinal and longitudinal directions respectively. The color bar depicts the spatial variation of aviation's $SO_2$ mass flux [kg m$^{-3}$ s$^{-1}$] according to the CMIP6 emissions inventory. The percentages represent the contribution (Eq. B1) of a grid cell to the total $SO_2$ mass flux.



The $SO_2$ mass flux contribution of grid cell $i$, $\Gamma_i^{SO_2}$, is given by Eq. B1:

$$\Gamma_i^{SO_2} = \frac{A_i \times SO_{2,i}}{\sum_j^T A_j \times SO_{2,j}} \times 100\%. \qquad (\text{Eq. B1})$$

The total number of grid cells is given by $T = \frac{180°}{\Delta lat} \times \frac{360°}{\Delta lon}$, $SO_{2,i}$ is the total mass within a grid cell $i$ and the area of a grid cell $A_i$ is calculated according to Kelly and Šavrič (2021):

$$A_i = R_\oplus^2 \times \Delta lat \times \Delta lon \times \cos \overline{lat_i}. \qquad (\text{Eq. B2})$$

$\overline{lat_i}$ is the average between the bounding latitudes of grid cell $i$ (i.e., its midpoint). The amount of $SO_2$ mass emitted in grid cell $i$, $SO_{2,i}$, is given by $SO_{2,i} = \Gamma_i^{SO_2} \times SO_{2,Total}$, where $SO_{2,Total}$ is approximately the total daily amount of $SO_2$ mass produced globally by aviation in 2015 according to CMIP6. The amount of $H_2SO_4$ is found by assuming that aviation's sulfur emissions are 98% $SO_2$ and 2% $H_2SO_4$. This means that the amount of $H_2SO_4$ mass emitted per grid cell, $H_2SO_{4,i}$, may be found in terms of $SO_{2,i}$, according to Eq. B3:

$$\frac{H_2SO_{4,i}}{H_2SO_{4,i} + \frac{M_{H_2SO_4}}{M_{SO_2}} \times SO_{2,i}} = 0.02 \Rightarrow H_2SO_{4,i} = \frac{0.02}{0.98} \times \frac{M_{H_2SO_4}}{M_{SO_2}} \times SO_{2,i}. \qquad (\text{Eq. B3})$$

The molecular masses of $SO_2$ and $H_2SO_4$ are denoted by $M_{SO_2}$ and $M_{H_2SO_4}$ respectively.

**Appendix C – tagging formulation (based on Grewe (2013)) of the coagulation process**

We derive the tagging formulations for the remaining eight aerosol modes by applying Eq. 9 to Eq. 6. For each mode, we first rewrite Eq. 6 into a more compact form using the notations of Eqs. 16a and 16b and as a function of the soluble mass fraction 945 $\theta$ relative to water. Each derivation fulfils the tagging additivity requirement of Eq. 10. As a reminder, we note that $x_i = x_i^{avi} + x_i^{rem}$ and $C_p = C_{NH_4,p} + C_{NO_3,p} + C_{Na,p} + C_{Cl,p} + C_{POM,p} + C_{BC,p} + C_{DU,p} + C_{H_2O,p}$.

Aitken mixed (km) mode:

$$\frac{\partial x_2}{\partial t}\bigg|_{coag} = F_2(\mathbf{x}) = \begin{cases} 0; \theta = 1 \text{ (Soluble)} \\ A\frac{x_1}{x_1 + C_1} - B\frac{x_2}{x_2 + C_2} + C\frac{x_3}{x_3 + C_3}; \theta \epsilon [0.1,1[ \text{ (Mixed)} \\ -B'\frac{x_2}{x_2 + C_2}; \theta \epsilon [0,0.1[ \text{ (Insoluble)} \end{cases} \qquad (\text{Eq. C1})$$





Here, the terms A, B, B′ and C are defined as:

$$A = \rho_1 \frac{\pi}{6} \left[ f_{1,2} + f_{1,3} \right],$$

$$B = \rho_2 \frac{\pi}{6} \left[ f_{2,4} + f_{2,5} + f_{2,6} + f_{2,7} + f_{2,8} \right],$$

$$B' = \rho_2 \frac{\pi}{6} \left[ f_{2,3} + f_{2,6} + f_{2,9} \right],$$

$$C = \rho_3 \frac{\pi}{6} f'_{2,3}.$$

Aviation's contribution is then given by:

$$\left. \frac{\partial x_2^{avi}}{\partial t} \right|_{coag} = \begin{cases} 0; \theta = 1 \text{ (Soluble)} \\ F_2(\mathbf{x}) \dfrac{AC_1 \dfrac{x_1^{avi}}{(x_1 + C_1)^2} - BC_2 \dfrac{x_2^{avi}}{(x_2 + C_2)^2} + CC_3 \dfrac{x_3^{avi}}{(x_3 + C_3)^2}}{AC_1 \dfrac{x_1}{(x_1 + C_1)^2} - BC_2 \dfrac{x_2}{(x_2 + C_2)^2} + CC_3 \dfrac{x_3}{(x_3 + C_3)^2}}; \theta \epsilon [0.1,1[ \text{ (Mixed)} \\ F_2(\mathbf{x}) \dfrac{x_2^{avi}}{x_2}; \theta \epsilon [0,0.1[ \text{ (Insoluble).} \end{cases} \quad \text{(Eq. C2)}$$

It is interesting to note that when $\theta = 1$ in Eq. C1, the coagulation tendency is zero as there is no collision event that produces a final particle in the soluble state when one of the colliding particles is in the km mode. This is consistent with Table 2 of Kaiser et al. (2014) in which all collision events are specified.

Aitken insoluble (ki) mode:

$$\left. \frac{\partial x_3}{\partial t} \right|_{coag} = F_3(\mathbf{x}) = \begin{cases} 0; \theta = 1 \text{ (Soluble)} \\ -C \dfrac{x_3}{x_3 + C_3}; \theta \epsilon [0.1,1[ \text{ (Mixed)} \\ A' \dfrac{x_1}{x_1 + C_1} + B' \dfrac{x_2}{x_2 + C_2} - C' \dfrac{x_3}{x_3 + C_3} + D' \dfrac{x_4}{x_4 + C_4} + E' \dfrac{x_5}{x_5 + C_5}; \theta \epsilon [0,0.1[ \text{ (Insoluble).} \end{cases} \quad \text{(Eq. C3)}$$

Here, the terms A, A′, B′, C′, D′ and E′ are defined as:

$$C = \rho_3 \frac{\pi}{6} \left[ f'_{1,3} + f'_{2,3} + f_{3,4} + f_{3,5} + f_{3,7} + f_{3,8} \right],$$

$$A' = \rho_1 \frac{\pi}{6} f_{1,3},$$





$$B' = \rho_2 \frac{\pi}{6} f_{2,3},$$

$$C' = \rho_3 \frac{\pi}{6} [f_{3,6} + f_{3,9}],$$

$$D' = \rho_4 \frac{\pi}{6} f'_{3,4},$$

$$E' = \rho_5 \frac{\pi}{6} f'_{3,5}.$$

Aviation's contribution is then given by:

$$\left.\frac{\partial x_3^{avi}}{\partial t}\right|_{coag} = \begin{cases} 0; \theta = 1 \text{ (Soluble)} \\ F_3(\mathbf{x}) \frac{x_3^{avi}}{x_3}; \theta \epsilon [0.1,1[ \text{ (Mixing)} \\ F_3(\mathbf{x}) \frac{A'C_1 \frac{x_1^{avi}}{(x_1+C_1)^2} + B'C_2 \frac{x_2^{avi}}{(x_2+C_2)^2} - C'C_3 \frac{x_3^{avi}}{(x_3+C_3)^2} + D'C_4 \frac{x_4^{avi}}{(x_4+C_4)^2} + E'C_5 \frac{x_5^{avi}}{(x_5+C_5)^2}}{A'C_1 \frac{x_1}{(x_1+C_1)^2} + B'C_2 \frac{x_2}{(x_2+C_2)^2} - C'C_3 \frac{x_3}{(x_3+C_3)^2} + D'C_4 \frac{x_4}{(x_4+C_4)^2} + E'C_5 \frac{x_5}{(x_5+C_5)^2}}; \theta \epsilon [0,0.1[ \text{ (Insoluble)}. \end{cases}$$
(Eq. C4)

Accumulation soluble (as) mode:

$$\left.\frac{\partial x_4}{\partial t}\right|_{coag} = F_4(\mathbf{x}) = \begin{cases} A \frac{x_1}{x_1+C_1} - D \frac{x_4}{x_4+C_4}; \theta = 1 \text{ (Soluble)} \\ -D' \frac{x_4}{x_4+C_4}; \theta \epsilon [0.1,1[ \text{ (Mixed)} \\ -D'' \frac{x_4}{x_4+C_4}; \theta \epsilon [0,0.1[ \text{ (Insoluble)}. \end{cases}$$
(Eq. C5)

Here, the terms $D'$ and $D''$ are defined as:

$$A = \rho_1 \frac{\pi}{6} f_{1,4},$$

$$D = \rho_4 \frac{\pi}{6} f_{4,7},$$

$$D' = \rho_4 \frac{\pi}{6} [f'_{2,4} + f'_{3,4} + f_{4,5} + f_{4,6} + f_{4,8} + f_{4,9}],$$

$$D'' = \rho_4 \frac{\pi}{6} [f'_{3,4} + f_{4,6} + f_{4,9}].$$





Aviation's contribution is given by:

$$\frac{\partial x_4^{avi}}{\partial t}\bigg|_{coag} = \begin{cases} F_4(\mathbf{x})\dfrac{AC_1\dfrac{x_1^{avi}}{(x_1+C_1)^2} - DC_4\dfrac{x_4^{avi}}{(x_4+C_4)^2}}{AC_1\dfrac{x_1}{(x_1+C_1)^2} - DC_4\dfrac{x_4}{(x_4+C_4)^2}}; \theta = 1 \text{ (Soluble)} \\[1em] F_4(\mathbf{x})\dfrac{x_4^{avi}}{x_4}; \theta\epsilon[0.1,1[ \text{ (Mixed)} \\[1em] F_4(\mathbf{x})\dfrac{x_4^{avi}}{x_4}; \theta\epsilon[0,0.1[ \text{ (Insoluble)}. \end{cases} \qquad (\text{Eq. C6})$$

Accumulation mixed (am) mode:

$$\frac{\partial x_5}{\partial t}\bigg|_{coag} = F_5(\mathbf{x}) = \begin{cases} 0; \theta = 1 \text{ (Soluble)} \\[0.5em] A'\dfrac{x_1}{x_1+C_1} + B'\dfrac{x_2}{x_2+C_2} + C'\dfrac{x_3}{x_3+C_3} + D'\dfrac{x_4}{x_4+C_4} - E'\dfrac{x_5}{x_5+C_5} + F'\dfrac{x_6}{x_6+C_6}; \theta\epsilon[0.1,1[ \text{ (Mixed)} \\[0.5em] -E''\dfrac{x_5}{x_5+C_5}; \theta\epsilon[0,0.1[ \text{ (Insoluble)}. \end{cases} \qquad (\text{Eq. C7})$$

Here, the terms $A'$, $B'$, $C'$, $D'$, $E'$, $F'$ and $E''$ are defined as:

$$A' = \rho_1\frac{\pi}{6}\big[f_{1,5} + f_{1,6}\big],$$

$$B' = \rho_2\frac{\pi}{6}\big[f_{2,4} + f_{2,5} + f_{2,6}\big],$$


$$C' = \rho_3\frac{\pi}{6}\big[f_{3,4} + f_{3,5}\big],$$

$$D' = \rho_4\frac{\pi}{6}\big[f'_{2,4} + f'_{3,4} + f_{4,5} + f_{4,6}\big],$$

$$E' = \rho_5\frac{\pi}{6}\big[f_{5,7} + f_{5,8} + f_{5,9}\big],$$

$$E'' = \rho_5\frac{\pi}{6}\big[f'_{3,5} + f_{5,6} + f_{5,9}\big],$$

$$F' = \rho_6\frac{\pi}{6}\big[f'_{1,6} + f'_{2,6} + f'_{4,6} + f'_{5,6}\big].$$






Aviation's contribution is given by:

$$\frac{\partial x_5^{avi}}{\partial t}\Big|_{coag} = \begin{cases} 0; \theta = 1 \text{ (Soluble)} \\ F_5(\mathbf{x})\frac{n}{d}; \theta\epsilon[0.1,1[ \text{ (Mixing)} \\ F_5(\mathbf{x})\frac{x_5^{avi}}{x_5}; \theta\epsilon[0,0.1[ \text{ (Insoluble).} \end{cases} \quad (\text{Eq. C8})$$

Here, n and d are abbreviations for the numerator and denominator:

$$n = A'C_1\frac{x_1^{avi}}{(x_1+C_1)^2} + B'C_2\frac{x_2^{avi}}{(x_2+C_2)^2} + C'C_3\frac{x_3^{avi}}{(x_3+C_3)^2} + D'C_4\frac{x_4^{avi}}{(x_4+C_4)^2} - E'C_5\frac{x_5^{avi}}{(x_5+C_5)^2} + F'C_6\frac{x_6^{avi}}{(x_6+C_6)^2},$$


$$d = A'C_1\frac{x_1}{(x_1+C_1)^2} + B'C_2\frac{x_2}{(x_2+C_2)^2} + C'C_3\frac{x_3}{(x_3+C_3)^2} + D'C_4\frac{x_4}{(x_4+C_4)^2} - E'C_5\frac{x_5}{(x_5+C_5)^2} + F'C_6\frac{x_6}{(x_6+C_6)^2}.$$

Accumulation insoluble (ai) mode:

$$\frac{\partial x_6}{\partial t}\Big|_{coag} = F_6(\mathbf{x}) = \begin{cases} 0; \theta = 1 \text{ (Soluble)} \\ -F\frac{x_6}{x_6+C_6}; \theta\epsilon[0.1,1[ \text{ (Mixed)} \\ A'\frac{x_1}{x_1+C_1} + B'\frac{x_2}{x_2+C_2} + C'\frac{x_3}{x_3+C_3} + D'\frac{x_4}{x_4+C_4} + E'\frac{x_5}{x_5+C_5} + \cdots \\ \cdots - F'\frac{x_6}{x_6+C_6} + G'\frac{x_7}{x_7+C_7} + H'\frac{x_8}{x_8+C_8}; \theta\epsilon[0,0.1[ \text{ (Insoluble).} \end{cases} \quad (\text{Eq. C9})$$

Here, the terms A', B', C', D', E', F', F, G', and H':

$$A' = \rho_1\frac{\pi}{6}f_{1,6},$$


$$B' = \rho_2\frac{\pi}{6}f_{2,6},$$

$$C' = \rho_3\frac{\pi}{6}f_{3,6},$$

$$D' = \rho_4\frac{\pi}{6}f_{4,6},$$

$$E' = \rho_5\frac{\pi}{6}f_{5,6},$$





$$F = \rho_6 \frac{\pi}{6} \left[ f'_{1,6} + f_{2,6} + f'_{4,6} + f'_{5,6} + f_{6,7} + f_{6,8} \right],$$

$$F' = \rho_6 \frac{\pi}{6} f_{6,9},$$

$$G' = \rho_7 \frac{\pi}{6} f'_{6,7},$$

$$H' = \rho_8 \frac{\pi}{6} f'_{6,8}.$$

Aviation's contribution is given by:

$$\left. \frac{\partial x_6^{avi}}{\partial t} \right|_{coag} = \begin{cases} 0 ; \theta = 1 \text{ (Soluble)} \\ F_6(\mathbf{x}) \dfrac{x_6^{avi}}{x_6} ; \theta \epsilon [0.1,1[ \text{ (Mixing)} \\ F_6(\mathbf{x}) \dfrac{n}{d} ; \theta \epsilon [0,0.1[ \text{ (Insoluble)} \end{cases} \tag{Eq. C10}$$

Here, n and d are abbreviations for the numerator and denominator:

$$n = A'C_1 \frac{x_1^{avi}}{(x_1 + C_1)^2} + B'C_2 \frac{x_2^{avi}}{(x_2 + C_2)^2} + C'C_3 \frac{x_3^{avi}}{(x_3 + C_3)^2} + D'C_4 \frac{x_4^{avi}}{(x_4 + C_4)^2} + E'C_5 \frac{x_5^{avi}}{(x_5 + C_5)^2} + \cdots$$

$$\cdots - F'C_6 \frac{x_6^{avi}}{(x_6 + C_6)^2} + G'C_7 \frac{x_7^{avi}}{(x_7 + C_7)^2} + H'C_8 \frac{x_8^{avi}}{(x_8 + C_8)^2}.$$

$$d = A'C_1 \frac{x_1}{(x_1 + C_1)^2} + B'C_2 \frac{x_2}{(x_2 + C_2)^2} + C'C_3 \frac{x_3}{(x_3 + C_3)^2} + D'C_4 \frac{x_4}{(x_4 + C_4)^2} + E'C_5 \frac{x_5}{(x_5 + C_5)^2} + \cdots$$

$$\cdots - F'C_6 \frac{x_6}{(x_6 + C_6)^2} + G'C_7 \frac{x_7}{(x_7 + C_7)^2} + H'C_8 \frac{x_8}{(x_8 + C_8)^2}.$$

Coarse soluble (cs) mode:

$$\left. \frac{\partial x_7}{\partial t} \right|_{coag} = F_7(\mathbf{x}) = \begin{cases} A \dfrac{x_1}{x_1 + C_1} + D \dfrac{x_4}{x_4 + C_4} ; \theta = 1 \text{ (Soluble)} \\ -G' \dfrac{x_7}{x_7 + C_7} ; \theta \epsilon [0.1,1[ \text{ (Mixed)} \\ -G'' \dfrac{x_7}{x_7 + C_7} ; \theta \epsilon [0,0.1[ \text{ (Insoluble)}. \end{cases} \tag{Eq. C11}$$





Here, the terms A, D, $G'$ and $G''$ are defined as:

$$A = \rho_1 \frac{\pi}{6} f_{1,7},$$


$$D = \rho_4 \frac{\pi}{6} f_{4,7},$$

$$G' = \rho_7 \frac{\pi}{6} \left[ f'_{2,7} + f'_{3,7} + f'_{5,7} + f'_{6,7} + f_{7,8} + f_{7,9} \right],$$

$$G'' = \rho_7 \frac{\pi}{6} \left[ f'_{6,7} + f_{7,9} \right].$$

Aviation's contribution is given by:

$$\left. \frac{\partial x_7^{avi}}{\partial t} \right|_{coag} = \begin{cases} F_7(\mathbf{x}) \dfrac{AC_1 \dfrac{x_1^{avi}}{(x_1 + C_1)^2} + DC_4 \dfrac{x_4^{avi}}{(x_4 + C_4)^2}}{AC_1 \dfrac{x_1}{(x_1 + C_1)^2} + DC_4 \dfrac{x_4}{(x_4 + C_4)^2}} ; \theta = 1 \text{ (Soluble)} \\ F_7(\mathbf{x}) \dfrac{x_7^{avi}}{x_7} ; \theta \epsilon [0.1,1[ \text{ (Mixed)} \\ F_7(\mathbf{x}) \dfrac{x_7^{avi}}{x_7} ; \theta \epsilon [0,0.1[ \text{ (Insoluble)}. \end{cases} \qquad \text{(Eq. C12)}$$

Coarse mixed (cm) mode:

$$\left. \frac{\partial x_8}{\partial t} \right|_{coag} = F_8(\mathbf{x}) = \begin{cases} 0 ; \theta = 1 \text{ (Soluble)} \\ A' \dfrac{x_1}{x_1 + C_1} + B' \dfrac{x_2}{x_2 + C_2} + C' \dfrac{x_3}{x_3 + C_3} + D' \dfrac{x_4}{x_4 + C_4} + E' \dfrac{x_5}{x_5 + C_5} + \cdots \\ \cdots + F' \dfrac{x_6}{x_6 + C_6} + G' \dfrac{x_7}{x_7 + C_7} + I' \dfrac{x_9}{x_9 + C_9} ; \theta \epsilon [0.1,1[ \text{ (Mixed)} \\ -H'' \dfrac{x_8}{x_8 + C_8} ; \theta \epsilon [0,0.1[ \text{ (Insoluble)}. \end{cases} \qquad \text{(Eq. C13)}$$

Here, the terms $A'$, $B'$, $C'$, $D'$, $E'$, $F'$, $G'$, $H''$ and $I'$ are defined as:

$$A' = \rho_1 \frac{\pi}{6} f_{1,8},$$

$$B' = \rho_2 \frac{\pi}{6} \left[ f_{2,7} + f_{2,8} \right],$$





$$C' = \rho_3 \frac{\pi}{6} \left[ f_{3,7} + f_{3,8} \right],$$

$$D' = \rho_4 \frac{\pi}{6} \left[ f_{4,8} + f_{4,9} \right],$$

$$E' = \rho_5 \frac{\pi}{6} \left[ f_{5,7} + f_{5,8} + f_{5,9} \right],$$

$$F' = \rho_6 \frac{\pi}{6} \left[ f_{6,7} + f_{6,8} \right],$$

$$G' = \rho_7 \frac{\pi}{6} \left[ f'_{2,7} + f'_{3,7} + f'_{5,7} + f'_{6,7} + f_{7,8} + f_{7,9} \right],$$

$$H'' = \rho_8 \frac{\pi}{6} \left[ f'_{6,8} + f_{8,9} \right],$$

$$I' = \rho_9 \frac{\pi}{6} \left[ f'_{4,9} + f'_{5,9} + f'_{7,9} + f'_{8,9} \right].$$

Aviation's contribution is given by:

$$\left. \frac{\partial x_8^{avi}}{\partial t} \right|_{coag} = \begin{cases} 0; \theta = 1 \ (\text{Soluble}) \\ F_8(\mathbf{x}) \dfrac{n}{d}; \theta \epsilon \left[ 0.1, 1 \right[ \ (\text{Mixing}) \\ F_8(\mathbf{x}) \dfrac{x_8^{avi}}{x_8}; \theta \epsilon \left[ 0, 0.1 \right[ \ (\text{Insoluble}). \end{cases} \qquad (\text{Eq. C14})$$

Here, n and d are abbreviations for the numerator and denominator:

$$n = A'C_1 \frac{x_1^{avi}}{(x_1 + C_1)^2} + B'C_2 \frac{x_2^{avi}}{(x_2 + C_2)^2} + C'C_3 \frac{x_3^{avi}}{(x_3 + C_3)^2} + D'C_4 \frac{x_4^{avi}}{(x_4 + C_4)^2} + E'C_5 \frac{x_5^{avi}}{(x_5 + C_5)^2} + \cdots$$

$$\cdots + F'C_6 \frac{x_6^{avi}}{(x_6 + C_6)^2} + G'C_7 \frac{x_7^{avi}}{(x_7 + C_7)^2} + I'C_9 \frac{x_9^{avi}}{(x_9 + C_9)^2}.$$

$$d = A'C_1 \frac{x_1}{(x_1 + C_1)^2} + B'C_2 \frac{x_2}{(x_2 + C_2)^2} + C'C_3 \frac{x_3}{(x_3 + C_3)^2} + D'C_4 \frac{x_4}{(x_4 + C_4)^2} + E'C_5 \frac{x_5}{(x_5 + C_5)^2} + \cdots$$

$$\cdots + F'C_6 \frac{x_6}{(x_6 + C_6)^2} + G'C_7 \frac{x_7}{(x_7 + C_7)^2} + I'C_9 \frac{x_9}{(x_9 + C_9)^2}.$$





Coarse insoluble (ci) mode:

$$\left.\frac{\partial x_9}{\partial t}\right|_{\text{coag}} = F_9(\mathbf{x}) = \begin{cases} 0; \theta = 1 \text{ (Soluble)} \\ -I' \dfrac{x_9}{x_9 + C_9}; \theta \epsilon [0.1,1[ \text{ (Mixed)} \\ A'' \dfrac{x_1}{x_1 + C_1} + B'' \dfrac{x_2}{x_2 + C_2} + C'' \dfrac{x_3}{x_3 + C_3} + D'' \dfrac{x_4}{x_4 + C_4} + E'' \dfrac{x_5}{x_5 + C_5} + \cdots \\ \cdots + F'' \dfrac{x_6}{x_6 + C_6} + G'' \dfrac{x_7}{x_7 + C_7} + H'' \dfrac{x_8}{x_8 + C_8}; \theta \epsilon [0,0.1[ \text{ (Insoluble)}. \end{cases} \quad \text{(Eq. C15)}$$

Here, the terms $A''$, $B''$, $C''$, $D''$, $E''$, $F''$, $G''$, $H''$ and $I'$ are defined as:

$$A'' = \rho_1 \frac{\pi}{6} f_{1,9},$$

$$B'' = \rho_2 \frac{\pi}{6} f_{2,9},$$


$$C'' = \rho_3 \frac{\pi}{6} f_{3,9},$$

$$D'' = \rho_4 \frac{\pi}{6} f_{4,9},$$

$$E'' = \rho_5 \frac{\pi}{6} f_{5,9},$$

$$F'' = \rho_6 \frac{\pi}{6} f_{6,9},$$

$$G'' = \rho_7 \frac{\pi}{6} f_{7,9},$$


$$H'' = \rho_8 \frac{\pi}{6} f_{8,9},$$

$$I' = \rho_9 \frac{\pi}{6} \left[ f'_{4,9} + f'_{5,9} + f'_{7,9} + f'_{8,9} \right].$$



Aviation's contribution is given by:

$$\frac{\partial x_9^{avi}}{\partial t}\bigg|_{coag} = \begin{cases} 0; \theta = 1 \text{ (Soluble)} \\ F_9(\mathbf{x})\dfrac{x_9^{avi}}{x_9}; \theta \epsilon\, [0.1,1[ \text{ (Mixed)} \\ F_9(\mathbf{x})\dfrac{n}{d}; \theta \epsilon\, [0,0.1[ \text{ (Insoluble).} \end{cases} \qquad \text{(Eq. C16)}$$

Here, n and d are abbreviations for the numerator and denominator:

$$n = A''C_1 \frac{x_1^{avi}}{(x_1 + C_1)^2} + B''C_2 \frac{x_2^{avi}}{(x_2 + C_2)^2} + C''C_3 \frac{x_3^{avi}}{(x_3 + C_3)^2} + D''C_4 \frac{x_4^{avi}}{(x_4 + C_4)^2} + E''C_5 \frac{x_5^{avi}}{(x_5 + C_5)^2} + \cdots$$

$$\cdots + F''C_6 \frac{x_6^{avi}}{(x_6 + C_6)^2} + G''C_7 \frac{x_7^{avi}}{(x_7 + C_7)^2} + H''C_8 \frac{x_8^{avi}}{(x_8 + C_8)^2}.$$

$$d = A''C_1 \frac{x_1}{(x_1 + C_1)^2} + B''C_2 \frac{x_2}{(x_2 + C_2)^2} + C''C_3 \frac{x_3}{(x_3 + C_3)^2} + D''C_4 \frac{x_4}{(x_4 + C_4)^2} + E''C_5 \frac{x_5}{(x_5 + C_5)^2} + \cdots$$

$$\cdots + F''C_6 \frac{x_6}{(x_6 + C_6)^2} + G''C_7 \frac{x_7}{(x_7 + C_7)^2} + H''C_8 \frac{x_8}{(x_8 + C_8)^2}.$$





**Appendix D – Location of liquid clouds according to satellite data**

Figure D1 plots the liquid cloud fraction (LCF; proportion of a grid cell area that is covered by liquid clouds) and the cloud

top pressure in hPa (CTP; pressure altitude of the uppermost region of a cloud) from the AVHRR-AMv3 satellite dataset for

both simulation periods: January – March, 2015 and July – September, 2015.

Figure D1 – Three-month means for the liquid cloud fractions (LCF) between (a) January – March of 2015 and (b) July – September of 2015. Panels (c) and (d) represent the same time periods as (a) and (b) respectively, but for the three-month means of the cloud top pressure (CTP) in hPa.

**Acknowledgements**

We would like to express our gratitude to Dr. Franziska Glassmeier from the Delft University of Technology for the fruitful

discussion regarding aerosol modeling.

This research was performed on the Dutch national e-infrastructure with the support of the SURF Cooperative under grant no.
EINF-5102/L2.



**Funding**

This research was part of the ACACIA (Advancing the Science for Aviation and Climate) Project, which was funded by the European Union under the Grant Agreement No. 875036.

**Code and Data Availability**

The Modular Earth Submodel System (MESSy) is continuously further developed and applied by a consortium of institutions.
The usage of MESSy and access to the source code is licenced to all affiliates of institutions which are members of the MESSy Consortium. Institutions can become a member of the MESSy Consortium by signing the MESSy Memorandum of Understanding. More information can be found on the MESSy Consortium Website at http://www.messy-interface.org (MESSy Submodels, 2024). The code presented here has been based on MESSy version 2.55.2 and will be available in the next official release (version 2.56) at https://doi.org/10.5281/zenodo.8360186 (The MESSy Consortium, 2025a).

The specific code version for AIRTRAC v2.0 that was used to generate the data presented in this article may be accessed via the following repository: https://doi.org/10.5281/zenodo.15965933 (The MESSy Consortium, 2025b).

The EMAC simulation output data that was produced and analyzed in this paper will be made openly available via a 4TU Data Archive at https://doi.org/10.4121/8d2cdb5f-b652-41db-95a2-5345b4c1e77c (Maruhashi et al., 2025b). This link will be minted on acceptance. In the meantime, the same output dataset is freely accessible via the following link:
https://data.4tu.nl/private_datasets/roLOC0VccLKAEsPosNgyhIFJrobb7JZqkZpUK8FWFQw (Maruhashi et al., 2025b).

**Supplement**

Supplementary figures and analyses regarding AIRTRAC v2.0 are also included in the same 4TU repository mentioned above.

**Author Contributions**

JM, MR, JH, VG and ID contributed to the conceptualization of the study. Coding of the new submodel was performed by JM,
with support from MR, PJ, VG and ID. JM, MR, VG and ID performed the analysis of the simulation output. Input files and initial setup for running EMAC with the MADE3 submodel were provided by MR. Mass conservation of aerosol tendencies (budget analysis) was verified by MS. JM produced the manuscript with input from all authors.

**Competing Interests**

Volker Grewe and Patrick Jöckel are editors for GMD.





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
