# Peer review of "AIRTRAC v2.0: a Lagrangian aerosol tagging submodel for the analysis of aviation SO4 transport patterns"

_EGUsphere, 2025_

## Author Comment (AC1)

Dear Editor, Referees,

We thank the referees for the insightful comments on our manuscript. Below, you will find our response to the referees, answering each comment and detailing how their suggestions were incorporated in the revision (referee comments in *italics*, our response **in bold** font, line numbers refer to the revised (clean) manuscript without tracked changes).

Overall, we have condensed the background information on aerosol direct and indirect radiative forcing, since quantifying these effects is not the focus of the manuscript. Instead, we provide additional detail on (i) the validity of the mathematical assumptions we adopt, (ii) sulfate transport patterns as a function of emission latitude and their likely interactions with liquid clouds, and (iii) our methodology.

Thank you again for considering our submission to GMD and we look forward to your response.

Sincerely,

Irene Dedoussi

**Referee #1 – Hongwei Sun**

*Review*

*This manuscript introduces a Lagrangian aerosol tagging submodel (AIRTRAC v2.0), which can help identify the aviation-emitted $SO_2$ and $H_2SO_4$ and track their contributions to $SO_4$ formation. AIRTRAC could be a useful tool for us to better evaluate sulfate emissions from aviation and estimate aviation's climatic impacts. I enjoyed reading the manuscript. It is well written and provides a thorough description, such as the detailed explanation of all terms contributing to the aerosol mass tendency in Eq. 2. I recommend a minor revision for the authors to address my comments below.*

**We would like to thank the referee for their time in providing constructive feedback that has improved our manuscript. We are also pleased to read that the reviewer enjoyed reading our manuscript and that AIRTRAC v2.0 is deemed a relevant and promising tool in improving our understanding of aviation-induced sulfate transport.**

*Major comments*

*Lines 552-553: The first-order Maclaurin polynomial is the linear approximation of the function f(x) near x=0, which cannot be used especially if x>k. To use the first-order Maclaurin polynomial, the authors at least need to show that x (avi term) is smaller than k (rem term).*

**Thank you for highlighting this point. In our original derivation we implicitly treated tagging ratios of the form $f(x) = \frac{x}{x+k}$ as convergent alternating geometric series, without explicitly proving the required convergence condition on the geometric ratio $r = \frac{x}{k}$. In other words, the series expansion is valid only when $\frac{x}{k} < 1$, thus, for the tagging ratios we require:**

$$f(x) = \frac{x}{x+k} = \sum_{n=1}^{\infty} (-1)^{n-1} \left(\frac{x}{k}\right)^n = \frac{x}{k} - \left(\frac{x}{k}\right)^2 + \left(\frac{x}{k}\right)^3 - \cdots, \text{if } \left|\frac{x}{k}\right| < 1. \qquad (\text{Eq. 1})$$

Eq. 1 clearly shows that if $\left|\frac{x}{k}\right| \geq 1$, the approximation $\frac{x}{x+k} \approx \frac{x}{k}$ quickly breaks down due to higher-order terms becoming dominant.

In our manuscript, $f(x)$ denotes tagging ratios of the form:

$$\frac{x}{x+k} \equiv \frac{C_j|^{avi}}{C_j|^{avi} + C_j|^{rem}}, j = \{SO_2, H_2SO_4, SO_4\}.$$

To justify the use of the first-order MacLaurin approximation, we compute

$$\left|\frac{x}{k}\right| \equiv \left|\frac{C_j|^{avi}}{C_j|^{rem}}\right|_{j=\{SO_2, H_2SO_4, SO_4\}}$$

for both the summer (July – September) and winter (January – March) simulations across the full EMAC model grid. Figures S7 through S12 of the Supplement depict the time series of these geometric ratios for each of the 28 emission points shown in Fig. 2 of the manuscript. Both numerator and denominator of $\left|\frac{x}{k}\right|$ are the area-weighted spatial means of the chemical species' volume mixing ratios computed following Eq. B2 of the manuscript.

As an example, we include Figure 1 below, which globally shows that $\left|\frac{x}{k}\right| < 1$ for $SO_2$ during a January – March simulation across all 28 emission points. The time series of the remaining species ($H_2SO_4$ and $SO_4$) and other simulation period also respect this condition and may be consulted in the Supplement. In the revised manuscript, we now explicitly state that this condition justifies using the first-order MacLaurin polynomials to linearize the tagging ratios (see lines 527 – 529). We have also complemented this section by explicitly mentioning that the linearization of the tagging ratio also applies to the aviation-attributable $SO_2$ (see lines 525 – 526).

[Figure]

**Figure 1 – Time series of the linearized tagging ratio** $\left|\dfrac{x}{k}\right| = \left|\dfrac{c_{SO_2}|^{avi}}{c_{SO_2}|^{rem}}\right|$ **for all 28 emission points (EP) for a simulation covering the January – March 2015 period.**

*Line 574: Why is there faster downward transport at point 10? My understanding is that because tropopause height is higher near tropics (point 10) than higher latitudes (point 8) [see Figure 2b in Sun et*

*al. (2023)], so emitted SO2 (at 240 hPa/10.6 km) at Point 8 may be located in the lower stratosphere, while emitted SO2 at Point 10 is definitely in the troposphere. I think this is worth mentioning, which can help to explain why "SO2 emitted at point 8 remains at higher altitudes for longer" (Line 580).*

*FYI: Sun, H., Bourguet, S., Eastham, S., & Keith, D. (2023). Optimizing injection locations relaxes altitude-lifetime trade-off for stratospheric aerosol injection. Geophysical Research Letters, 50, e2023GL105371. https://doi.org/10.1029/2023GL105371 (already in your references).*

**We thank the referee for this insightful comment. The climatological tropopause is indeed relevant for the vertical transport of emitted species, as it demarcates the relatively stable stratosphere, where vertical mixing is weak and descent is typically slow, from the troposphere, which is characterized by stronger turbulence and convective overturning. In response, we have updated Figs. 5 to 7 of the revised manuscript (previously Figs. 4 to 6) to include the climatological tropopause pressure altitude, shown as a dotted black line.**

**Figure 2, reproduced from Fig. 5 of the revised manuscript, illustrates that a substantial fraction of SO₂ emitted at emission point 8 is injected at or above the climatological tropopause, that is, in the lower stratosphere, whereas emissions at point 10 at the same pressure altitude occur well within the troposphere. This distinction could provide a consistent explanation for the more rapid downward transport and shorter lifetimes at point 10 (e-folding lifetime $\tau = 17$ days), and conversely for the longer residence times when injection occurs in the lower stratosphere at point 8 (e-folding lifetime $\tau = 27$ days).**

[Figure]

**Figure 2 – Figure 5 of the revised manuscript with climatological tropopause (dotted black line) shown in panels (c) and (d).**

**We have added this explanation to the manuscript (see lines 554 – 559).**

Additionally, we now highlight in the Conclusion, the importance of emission latitude in influencing the vertical transport behavior and species lifetime (see lines 856 – 860).

Lastly, we now also discuss this important link between emission latitude, climatological tropopause and $SO_4$ lifetimes in the abstract (see lines 27 – 29).

*Section 4.3: All the analyses of seasonal effects (Figure 7) are based on a one-year emission scenario (2015). Without using the climatological mean, it is possible that the difference between winter and summer is caused by other perturbations rather than the seasonal cycle. The possible perturbation includes internal (e.g., 2015 is a strong El Niño year) or external (e.g., volcano eruptions) forcing.*

Thank you for highlighting this point. We agree that multi-year simulations would be required to attribute the differences shown in Fig. 7 (now Fig. 8 in the revised manuscript) to the climatological seasonal cycle. The primary aim of this manuscript is to introduce and describe the new Lagrangian aerosol tagging submodel, and Section 4.3 is intended to demonstrate that AIRTRAC v2.0 can be used to diagnose sensitivity to time-varying background conditions across seasons within the simulated year, including changes in solar input and background chemistry.

To make this scope explicit, we have retitled Section 4.3 to "Intra-annual variability" and mention in the Fig. 8 caption that an "intra-annual comparison across different seasons" is being shown. We also now state explicitly that since Fig. 8 results are based on a single simulation year, their cross-seasonal differences should not be attributed to the climatological seasonal cycle alone, since interannual anomalies may also contribute to the contrasts shown. As additional context, we have added Paek et al. (2017) to note that 2015 to 2016 coincided with an extreme El Niño event.

We have added this clarification in lines 654 – 658.

Additionally, in line 534 at the beginning of Section 4, we reword a sentence to clarify that the impacts are across seasons and not necessary caused by the seasonal shift itself. Previously, we had stated that "The impact of seasonal shifts in the lifetimes of $SO_2$, $SO_4$, …, is also considered". We now write:

"The shifts in the lifetimes of $SO_2$, $SO_4$ and in the mean productive efficiency of $SO_4$ across seasons is also considered."

Lastly, we update the Conclusion so that we are not attributing seasonality as the sole driver behind intra-annual shifts in lifetime and production efficiencies (see lines 868 – 872).

Paek, H., Yu, J.-Y. and Qian, C.: Why were the 2015/2016 and 1997/1998 extreme El Niños different?, Geophys. Res. Lett., 44, 1848 – 1856, https://doi.org/10.1002/2016GL071515, 2017.

*Line 723: Besides the Northern Atlantic and Southern Tropics, there are several other stratocumulus decks, such as the Northeast Pacific, as shown in Figure D1 (b) and (d). Therefore, aviation sulfate from Point 8 is able to interact with liquid clouds in the Northeast Pacific. See Figure 2 in Muhlbauer et al. (2014).*

*FYI: Muhlbauer, A., McCoy, I. L., and Wood, R.: Climatology of stratocumulus cloud morphologies: microphysical properties and radiative effects, Atmos. Chem. Phys., 14, 6695–6716, https://doi.org/10.5194/acp-14-6695-2014, 2014.*

We agree that emission point 8 in Fig. 8 (now Fig. 9 in the revised manuscript) is also likely to intersect the Northeast Pacific liquid cloud region identified in our Figs. D1b and D1d, and in Muhlbauer et al. (2014). The purpose of Fig. 9, however, is to highlight how sulfate transport pathways differ with emission latitude by using the liquid clouds in the Southern Tropical Belt as a test case. In our context,

the conclusion remains unchanged: the air parcel from emission point 10 is transported into the Southern Tropical Belt liquid cloud region, whereas the air parcel from emission point 8 is not.

In the original submission we also noted that the air parcel from point 8 may interact with liquid clouds outside the Southern Tropical Belt, for example in the North Atlantic (lines 734 to 735 of the originally submitted manuscript): "Although it is unlikely that $SO_4$ will reach the primary region with liquid clouds indicated in blue, it could still interact with some clouds present in the North Atlantic (see Fig. D1)." Still, we agree that it is useful to extend this clarification to include the North Pacific liquid cloud structure as well, and we have added a clarification in lines 725 – 727.

We have also revised the caption of Fig. 9 from the manuscript to make clear that the analysis focuses specifically on transport into the Southern Tropical Belt. The updated caption may be found in lines 730 – 734.

We have also included the suggested reference to further emphasize the consistency between the global liquid cloud structures represented across several studies and datasets (see lines 715 – 717).

Lastly, the Conclusion also now clarifies the region of analysis for aerosol-cloud interactions (see lines 876 – 880).

*Minor comments:*

*Line 203-204: How many vertical layers are in the upper troposphere and low stratosphere? I think you should have more vertical levels in the free troposphere than in the boundary layer.*

The 41 vertical model layers of the T42L41 configuration are shown in Fig. 3 below for July to September. Panel 3a displays the zonal and time mean hybrid mid-level pressure layers together with the zonal and time mean climatological tropopause, illustrating that most model layers reside in the troposphere. Panel 3b quantifies the number of layers located above and below the climatological tropopause and also reports the number of layers between the upper troposphere and lower stratosphere (UTLS), which we approximate here as the 100-to-400-hPa layer. The partitioning varies with latitude: poleward of about 60°, more layers lie above the tropopause than below, whereas in the midlatitudes and Tropics most layers remain in the troposphere. At the Equator, for example, 33 layers are below the tropopause and 8 are above. The number of layers between 100 and 400 hPa is approximately 16 and remains nearly constant with latitude.

[Figure]

**Figure 3 – (a) Vertical structure of the EMAC T42L41 grid relative to the climatological tropopause. (a) Zonal and time mean pressure altitudes at hybrid layer midpoints (curves in blue) for the period July – September, 2015. The climatological tropopause is represented as a thick orange line. (b) Corresponding number of model layers below and above the climatological tropopause. The upper troposphere and lower stratosphere (UTLS) are assumed to be approximately between 100 – 400 hPa.**

We have added this two-panel figure to the Supplement and included additional text in the manuscript describing the vertical distribution of model layers above and below the tropopause (see lines 173 – 176).

*Line 210-211: Because 28 emission points are at different latitudes, why are they all at the same altitude, especially if we consider the fact that tropopause height varies largely at different latitudes? Would consider height variation make the emissions points more realistic (Line 222-223).*

The emission altitude was selected by identifying the pressure level at which the zonal mean aviation $SO_2$ mass flux reaches its maximum (Fig. 3 in the revised manuscript). This level therefore corresponds to the pressure altitude where aircraft operations and associated $SO_2$ injection are greatest according to the CMIP6 emissions inventory. We further prescribe the same emission altitude for all emission points to isolate the role of latitude and to enable a consistent comparison of transport behaviour along the main flight corridors. A natural next step is of course to assess sensitivity to emission altitude by repeating the experiments at levels above and below the chosen 238.2 hPa.

*Line 226-228: the altitude is found by the zonal-mean max of SO2 mass flux, which should be a function of latitude.*

Correct, the zonally averaged $SO_2$ mass flux becomes a function of latitude and altitude once it has been averaged across the longitude and time dimensions. Figure 3 in the revised manuscript plots this variable as a function of altitude and latitude and the maximum $SO_2$ mass flux occurs at an altitude of around 240 hPa. In the manuscript, we have added a minor change to emphasize this (see lines 199 – 200).

*Line 505: definition of A, A', and A'' needs more explanation. What's the meaning of all terms on the right-hand side of the equations (e.g., $f_{1,4}$)? Are they all constants*

**The coefficients $A$, $A'$ and $A''$ are not universal constants as they depend on the sizes of the colliding aerosols and may also depend on other thermodynamic variables like the atmospheric temperature and pressure (see Eqn. C17 of the revised manuscript) and on the mean free path $\lambda$. However, they are assumed to be independent of the state variables $x_1, x_2, \ldots, x_9$. We had already defined the right-hand side of $f_{p,q}$ earlier in Eqns. 16a and 16b of the manuscript. Additionally, the individual terms of Eq. 16 had also already been defined previously at the end of Section 2.3.3 (see lines 327 – 330).**

**We agree, however, that without explicitly stating the mathematical form of the Brownian coagulation kernels $\beta$, it may not be clear which variables ultimately control the A coefficients and the $f_{p,q}$ terms. We have therefore added the definitions of $\beta$ (following Whitby et al., 1991) to Appendix C, and we have expanded the discussion in Section 2.3.3 to clarify the variables on which $\beta$ depends (see lines 327 – 331).**

**We have added the following explanation regarding the mathematical formulations of $\beta$ for the different aerosol regimes in Appendix C (see lines 1057 – 1066).**

**Whitby, E. R., McMurray, P., Shankar, U., and Binkowski, F.: Modal Aerosol Dynamics Modeling, Tech. Rep. 600/3-91/020, Atmospheric Research and Exposure Assess. Lab., US Environmental Protection Agency, Research Triangle Park, available as NTIS PB91-161729/AS, 1991.**

*Line 553: typo: "McLaurin" should be "Maclaurin".*

**The typo has been fixed (see line 525).**

**Anonymous Referee #2**

*This study introduced a Lagrangian aerosol tagging submodel for sulfate aerosols into the atmospheric chemistry modelling framework. Compared to the old version, an important advancement is the incorporation of aerosol microphysical processes in the tagging module, which can enable better quantification of sulfate transport patterns. Overall, this manuscript is well-written and -organized. I recommend some revisions before the publication.*

**We would like to thank the reviewer for the helpful feedback provided and for considering AIRTRAC v2.0 as an "important advancement" relative to its predecessor.**

*First, I do suggest the authors to shorten the length the manuscript. Some content can be moved into Supplementary Materials. For example, you don't need to give too detailed background on aerosol effects on cloud albedo, as this is not the key focus of your study. And please refine the methods so that the core parts can be kept in the main text and shown to readers. Describing the new development of your model is interesting enough.*

**We agree that the background section can be streamlined. Following the reviewer's recommendation, we have substantially condensed the discussion of aerosol indirect effects and associated radiative forcing in the Introduction by moving Table 1 (now Table S1), together with the three accompanying paragraphs (formerly lines 102 – 119 in original submission) that summarize the studies cited in the table, to the Supplement. The corresponding references have been removed from the main reference list. Overall, these changes have reduced the manuscript length by approximately two pages.**

The paragraph that previously described the Twomey and Albrecht effects in detail has now been significantly shortened to retain only the essential points of aerosol-cloud interactions (see lines 74 – 80).

To refine the presentation of our methodology, we have expanded Section 2.2 to better justify the placement of the emission points. In particular, we now describe in more detail the zonal mean aviation $SO_2$ mass flux and how it served as the guiding metric for selecting a common emission altitude for the 28 points. The corresponding latitude–pressure altitude cross section (formerly Fig. B1 of the manuscript) has been moved from Appendix B to the main text (now Figure 3 of the revised manuscript). We have also checked, using the Coblis simulator, that its color scheme remains clear and interpretable for all color-blindness settings. We consider the detailed description of the auxiliary EPS_selector tool to be secondary to the AIRTRAC submodel itself, and it is therefore retained in Appendix B. Finally, for the coagulation tagging formulation (Section 3.4), the manuscript already provides a fully worked derivation for the soluble Aitken mode; the remaining derivations follow the same rationale and can therefore remain in Appendix C.

*In Section 4.1, please explain why the emission points 8 and 10 are selected for analyzing. How the result could be changed regarding other emission points, e.g., over Pacific Ocean?*

We have chosen emission points 8 and 10 as our test cases because they exhibit distinct vertical transport patterns, which will have different implications for potential aerosol-cloud interactions. Emission point 8, for instance, exhibits longer residence time in the vicinity of the emission region while emission point 10 quickly descends and has a shorter lifetime. Upon plotting the climatological tropopause as suggested by the other reviewer, we now are better able to understand that this selection has meant choosing an emission point entirely within the troposphere (emission point 10) and another partly in the lower stratosphere (emission point 8). Another reason is that if we focus on the Southern Tropical liquid cloud belt, these emission points are also great examples to see how emission latitude may greatly impact the likelihood of aerosol interaction with a certain deck of liquid clouds. We have included a brief explanation at the end of Section 4.1 regarding the selection of these emission points (see lines 544 – 546).

It is of course possible to choose other emission point combinations that would yield interesting results as well. However, the core objective of this manuscript is to exemplify the usefulness of AIRTRAC v2.0 using one test case and not perform an in-depth global analysis of transport patterns. The $SO_2$, $H_2SO_4$ and $SO_4$ vertical profiles (pressure altitude vs. latitude) for all emission points and both simulation periods, however, have been added to the Supplement for the interested reader (Figs. S14 to S19). The $SO_2$ profiles for July – September (Fig. 4) are shown in the next page as an example.

**Figure 4 – Vertical profiles of SO₂ for all 28 emission points during the July – September simulation. Green triangle indicates the approximate emission location and the dotted black line represents the climatological tropopause.**

*Please indicate a and b in Figure 8 panel.*

**We have added these letters to the panel plot of Fig. 9 (previously Fig. 8) of the manuscript.**

*I would like to give further comments on a revised version.*

We thank the reviewer for their constructive feedback and for their willingness to review the revised manuscript. We have addressed all the points raised and believe the clarifications and improvements have significantly strengthened the paper. We look forward to any further comments on this updated version.

Notification to Authors

*1. Please ensure that the colour schemes used in your maps and charts allow readers with colour vision deficiencies to correctly interpret your findings. Please check your figures using the Coblis – Color Blindness Simulator (https://www.color-blindness.com/coblis-color-blindness-simulator/) and revise the colour schemes accordingly with the next file upload request. -> Fig. 4, 5, 6, 8*

The color schemes of Figures 4, 5, 6, and 8 of the manuscript (now renumbered to Figs. 5, 6, 7 and 9 in the revised manuscript) were changed to a more colorblind-friendly "Blues" palette and evaluated using the Coblis simulator to ensure their clear interpretability. Although Figure 10 (previously Fig. 9) was not mentioned above and it does not apply the same "Blues" palette, we have nonetheless checked that it is safely and unambiguously interpretable with its original color scheme across the various Coblis color-blindness options.

*2. A "Short summary" system section contains scientific abbreviations. Please be aware that scientific abbreviations (excluding chemical elements) must have their full written explanations. However, do not forget that there is a limit to characters (not words!) for "Short summary": it must be < 500 characters.*

Please find below our Short Summary with the acronym "EMAC" now defined:

Aerosol-cloud interactions remain a large source of uncertainty in assessing aviation's climate impact. We develop, evaluate and present AIRTRAC v2.0 within the ECHAM-MESSy Atmospheric Chemistry (EMAC) modeling framework, which tracks aviation-emitted $SO_2$ and $H_2SO_4$ as they are chemically transformed into $SO_4$ aerosols and transported in the atmosphere. The development allows the identification of atmospheric regions with elevated potential for aerosol–cloud interactions due to $SO_4$ from aircraft.